# Gestational diabetes augments group B *Streptococcus* infection by disrupting maternal immunity and the vaginal microbiota

Vicki Mercado-Evans[1,2], Marlyd E. Mejia[1], Jacob J. Zulk[1], Samantha Ottinger[1], Zainab A. Hameed[1], Camille Serchejian[1], Madelynn G. Marunde[1], Clare M. Robertson[1], Mallory B. Ballard[1], Simone H. Ruano[3], Natalia Korotkova [4,5], Anthony R. Flores[6], Kathleen A. Pennington[3] & Kathryn A. Patras [1,7] ✉

Group B *Streptococcus* (GBS) is a pervasive perinatal pathogen, yet factors driving GBS dissemination *in utero* are poorly defined. Gestational *diabetes mellitus* (GDM), a complication marked by dysregulated immunity and maternal microbial dysbiosis, increases risk for GBS perinatal disease. Using a murine GDM model of GBS colonization and perinatal transmission, we find that GDM mice display greater GBS *in utero* dissemination and subsequently worse neonatal outcomes. Dual-RNA sequencing reveals differential GBS adaptation to the GDM reproductive tract, including a putative glycosyl-transferase (*yfhO*), and altered host responses. GDM immune disruptions include reduced uterine natural killer cell activation, impaired recruitment to placentae, and altered maternofetal cytokines. Lastly, we observe distinct vaginal microbial taxa associated with GDM status and GBS invasive disease status. Here, we show a model of GBS dissemination in GDM hosts that reca-pitulates several clinical aspects and identifies multiple host and bacterial drivers of GBS perinatal disease.

Group B *Streptococcus* (GBS, *Streptococcus agalactiae*) is a leading agent of neonatal morbidity and mortality globally and is responsible for ~150,000 stillbirths or infant deaths annually[1,2]. GBS asymptoma-tically colonizes the vagina of ~18% of pregnant women[3], and subse-quently ~20 million infants are exposed to maternal GBS at, or before, time of delivery[1,2]. In the absence of intrapartum antibiotic prophy-laxis, about half of infants born to GBS-positive women become colonized and a subset (~2%) proceed to develop invasive infection[4]. Clinical data supports that GBS can invade the uterus prior to disease onset or stillbirth, even with membranes intact[5–8], yet the factors driving GBS ascension into the uterus are poorly defined. The current U.S. standard of care, antibiotic prophylaxis to GBS-positive mothers during labor, fails to prevent GBS-associated stillbirths or preterm births, or late onset GBS disease, and exposes ~1 million U.S. infants to

[1]Department of Molecular Virology and Microbiology, Baylor College of Medicine, Houston, TX 77030, USA. [2]Medical Scientist Training Program, Baylor College of Medicine, Houston, TX 77030, USA. [3]Department of Obstetrics and Gynecology, Baylor College of Medicine, Houston, TX 77030, USA. [4]Department of Microbiology, Immunology and Molecular Genetics, University of Kentucky, Lexington, KY, USA. [5]Department of Molecular and Cellular Biochemistry, University of Kentucky, Lexington, KY, USA. [6]Division of Infectious Diseases, Department of Pediatrics, McGovern Medical School, UTHealth Houston, Children's Memorial Hermann Hospital, Houston, TX, USA. [7]Alkek Center for Metagenomics and Microbiome Research, Baylor College of Medicine, Houston, TX 77030, USA. ✉e-mail: katy.patras@bcm.edu

antibiotics each year[9]. This exposure has consequences on the developing infant microbiota including a rise in the frequency of antimicrobial resistance[10]. Understanding the biological principles controlling GBS-host dynamics is critical to developing defined, long-lasting preventions for GBS infections in pregnancy and the early neonatal period.

GBS possesses an arsenal of virulence factors (e.g., β-hemolysin/cytolysin, adhesins, hyaluronidase)[11–15] and environment-sensing two component regulatory systems (e.g., CovRS, SaeRS)[11,16,17] that facilitate colonization of the vaginal mucosa and/or invasion of host reproductive tissues. Transcriptomic analyses have revealed that GBS optimizes expression of virulence and metabolic genes to adapt to physiologically relevant stimuli such as pH and glucose in vitro[18,19] and human fluids including blood and amniotic fluid ex vivo[20–22]. GBS isolated from the non-pregnant murine vagina differentially expresses adhesins and metabolic genes compared to GBS grown in bacteriologic media. Further, using transposon mutagenesis, investigators have identified critical GBS factors for murine uterine ascension, yet it is unknown how GBS transcriptionally adapts as it ascends the vaginal tract to the uterus during pregnancy[17,23].

Maternal metabolic disorders including obesity and gestational diabetes mellitus (GDM) are associated with a 1.4 to 3.1-fold increased risk for maternal GBS colonization[24,25] and a 3 to 5-fold increased risk for GBS maternal and neonatal sepsis[25,26]. This clinical evidence implies the maternal diabetic environment is altered in favor of GBS colonization and dissemination, but the factors driving this phenomenon are unknown. Spontaneous hyperglycemia and insulin intolerance are pathophysiological hallmarks of GDM[27] and are frequently accompanied by maternal systemic immune dysregulation. Most clinical studies focused on maternal peripheral immune profiles and serum biomarkers. Observed differences include increased circulating T helper cell subsets and NK cells in GDM patients with a shift from anti-inflammatory to pro-inflammatory profiles[28–31] in tandem with increased serum inflammatory mediators and cytokines[32–34]. Although increased placental macrophages and neutrophils are observed in GDM pregnancies[35,36], GDM perturbations to reproductive immunity are not well-characterized. Furthermore, GDM is accompanied by altered maternal fecal and vaginal microbiome compositions which may further aggravate mucosal dysfunction[37–39]. These clinical findings suggest discordant host-microbe interactions in the GDM mucosa may provide an advantage to pathogens, such as GBS, over commensal microbes.

Here, we show that a diet-induced gestational diabetic model mirrors the pathophysiology of GBS disease in GDM, specifically through aberrant maternal immunity, differential GBS transcriptional adaptation, and altered composition of the vaginal microbiota during pregnancy. Considering the altered metabolic, immune, and microbial profiles in gestational diabetes compared to non-diabetic pregnancy, we hypothesized that GDM perturbations to host physiology contribute to increased host susceptibility and augmented GBS virulence. To test this hypothesis, we developed a mouse model of GBS vaginal colonization and the natural course of maternal-to-offspring transmission in a gestational diabetic host. Using this model, we performed integrative analysis of GBS burden and transmission, host and GBS global transcriptional profiling, reproductive tract cytokine and immune cell profiles, and longitudinal 16S rRNA amplicon sequencing of the vaginal microbiota during pregnancy. These findings advance our current mechanistic viewpoint on host and GBS factors that drive GBS disease in pregnancies impacted by diabetes.

## Results

### Gestational diabetes increases GBS dissemination to fetal tissues
We adapted a previously described diet-induced model of gestational diabetes in C57BL/6J mice[40–42]. Mice are started on either a high-fat high-sucrose (GDM group), or low-fat no-sucrose (pregnant control group) diet one week before mating and maintained on these diets until the experimental endpoint (Fig. 1a). In this model, mice develop multiple GDM-like symptoms during pregnancy, including glucose intolerance upon glucose challenge (Fig. 1b), decreased serum insulin paired with lower total beta cells, insulin resistance, and maternal dyslipidemia by embryonic day 13.5 (E13.5)[40–42]. In line with prior studies[40], we confirmed glucose intolerance was specific to pregnancy and not observed one day before mating, nor was maternal weight impacted by diet (Supplementary Fig. 1a, b). To model mid-gestational GBS vaginal colonization, mice were vaginally inoculated with GBS strain A909 (serotype Ia) on E14.5 and E15.5, and maternal and fetal tissues were collected on E17.5 before parturition (potential range of E15.5–17.5 based on mating scheme as described in Methods) (Fig. 1a). GDM mice had similar GBS burdens throughout the lower and upper reproductive tract compared to controls (Fig. 1c), and similar glucose levels in vaginal, uterine and placental tissues (Supplementary Fig. 1c). To distinguish diet vs. pregnancy-mediated effects, non-pregnant mice on either diet were included in these studies (Supplementary Fig. 1d). Pregnant mice on the HFHS diet (GDM group) had significantly greater vaginal and cervical GBS burden compared to non-pregnant mice on the HFHS diet, with no pregnancy-related differences for mice on the control diet (Supplementary Fig. 1e). Additionally, non-pregnant mice on the HFHS diet had significantly decreased GBS burdens in vaginal and cervical tissues compared to non-pregnant mice on control diet (Supplementary Fig. 1e) These findings suggest that our observations in GDM mice are not solely explained by diet or pregnancy alone, but rather the complex biological interplay of diet and pregnancy in this model.

Incidence of adverse events including maternal bacteremia and fetal resorption (~30% and ~40% of pregnancies respectively) were not significantly different between groups (Fig. 1d, e). Consistent with observations in uninfected mice[41], GDM fetuses were significantly larger on E17.5 despite GBS challenge (Fig. 1f). Fetuses from GDM dams were significantly more likely to have GBS positive placental and liver tissues (Fig. 1g) and had higher GBS burdens in these tissues (Fig. 1h) compared to fetuses from control dams. We hypothesized that GDM disrupts placental integrity and thus would be more permissive of GBS dissemination to fetal tissues; however, placental containment, defined as GBS detection in the placenta without translocation to respective fetal liver, was not different between groups (Fig. 1i). Disaggregation by fetal sex revealed that females in the control group had significantly greater invasive systemic GBS compared to males determined by liver bacterial burdens, with no sex-specific differences observed in the GDM cohort (Fig. 1j, k).

We also evaluated GBS strain CNCTC10/84 (serotype V), a hypervirulent, hyperhemolytic strain driven by decreased expression of the CovRS (control of virulence) two component system[43]. Similar to A909, CNCTC 10/84 displayed a trend towards higher vaginal GBS burdens in GDM mice (Supplementary Fig. 2a), but no differences in incidence of bacterial uterine ascension nor fetal demise were observed (Supplementary Fig. 2b–d). As with A909, CNCTC 10/84 was significantly more likely to be detected, and at greater amounts, in GDM placentae and fetal livers (Supplementary Fig. 2e, f). CNCTC 10/84 confinement to the placenta was decreased in GDM dams suggesting enhanced invasive potential of this strain (Supplementary Fig. 2g). Given that multiple aspects of GBS infection in the gestational diabetic host were conserved across two GBS strains, A909 was used for the remainder of the study.

### Gestational diabetes alters GBS transcription across tissues
We performed dual RNA-sequencing of murine vaginal, uterine and placental tissues and GBS harbored in each tissue on E17.5, with three objectives: (A) To identify GBS genes important for persistence in the gravid uterus, (B) identify genes important for GBS placental invasion, and (C) characterize GBS adaptation to a gestational diabetic

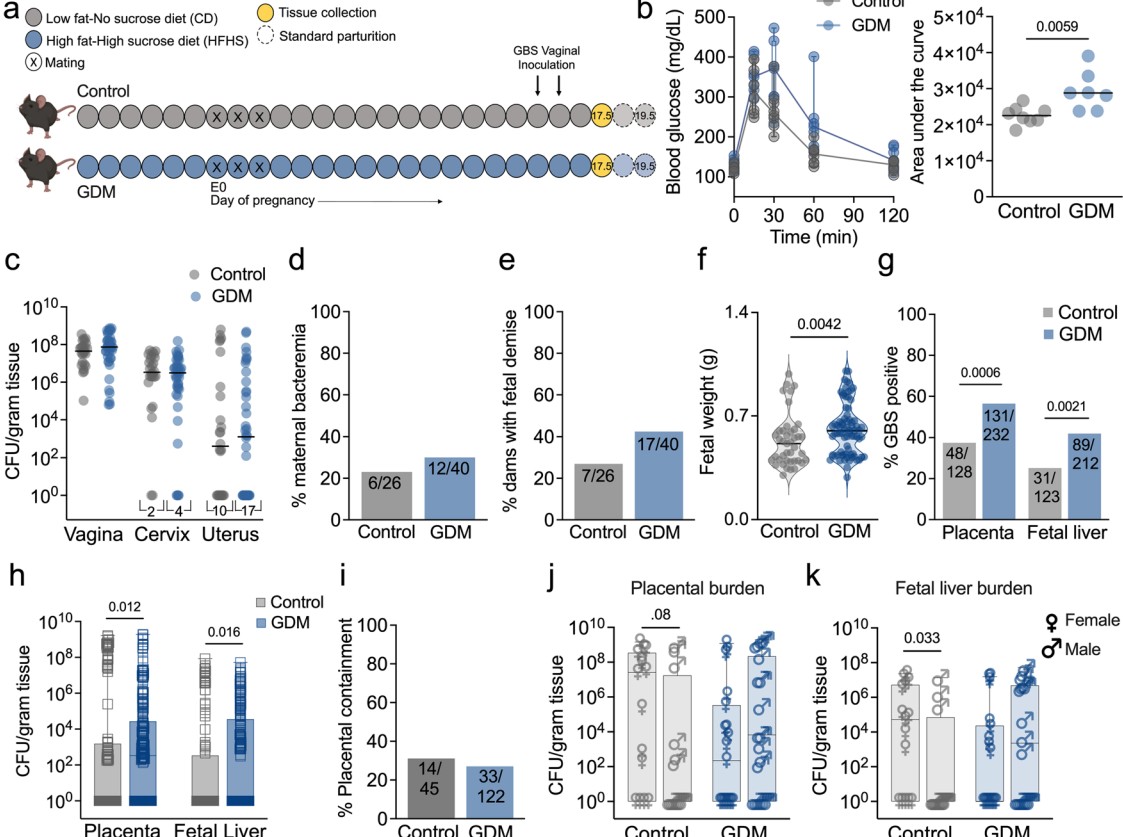

**Fig. 1 | Gestational diabetes enhances *in utero* group B Streptococcal fetal invasion in a murine model of ascending infection. a** Experimental timeline for GDM induction via a high-fat high-sucrose (HFHS) diet followed by mid-gestational GBS vaginal colonization, and tissue collection on E17.5. Mouse image created with BioRender.com. **b** Blood glucose concentration and area under the curve during glucose tolerance test on E13.5. **c** GBS burden in maternal reproductive tract tissues. Proportion of dams with (**d**) bacteremia (detectable CFU) or (**e**) fetal reabsorptions. **f** Fetal weights from control and GDM dams. **g** Percentage of placentae and fetal livers that were GBS positive (detectable CFU) and (**h**) corresponding GBS burdens. **i** Percentage of placental-fetal units that had GBS detected in the placenta with no detection in the corresponding fetal liver. **j** Placental and (**k**) fetal liver burdens stratified by fetal sex for a randomly selected subset. Data represent 2 independent (**b**) and 5 independent replicates (**c–k**). Curves represent medians and error bars are the interquartile ranges (**b**). Points represent individual samples and lines indicate medians (**b, c, f**). Box and whisker plots extend from 25th to 75th percentiles and show all points (**g, j, k**). $n = 26$ control and 40 GDM dams (**c–e**), $n = 41$ control and 75 GDM fetuses (**f**), $n = 128$ control and 232 GDM placentae, 123 control and 212 GDM fetal livers (**g, h**), $n = 21$ control (15 female, 14 male) and $n = 30$ GDM (19 female, 19 male) paired samples (**j, k**). Source data are provided as a Source Data file. Data was analyzed by two-tailed Mann–Whitney *t* test (**b, c, f, h, j, k**) and two-sided Fisher's exact test (**d, e, g, i**).

host. Principal component analysis revealed that GBS displays tissue-specific transcriptional clustering independent of host diabetic status (Fig. 2a). Vaginal and uterine RNA-seq was performed in two independent batches with some variation in GBS transcriptional profiles occurring between batches (Supplementary Table 1). When comparing uterine GBS vs. vaginal GBS transcriptional profiles, within pregnant controls and GDM groups, we identified differentially expressed genes (DEGs) contributing to tissue-specific signatures (Fig. 2b). Of the 15 DEGs in uterine GBS, 12 were shared in both control and GDM groups and included upregulation of genes responsible for ribose transport and carbohydrate metabolism and downregulation of a glutamic endopeptidase and several uncharacterized genes (Fig. 2c–e, Supplementary Table 1). Two DEGs were uniquely regulated in GDM mice and one was uniquely regulated in control mice (Fig. 2c–e, Supplementary Table 1). The two genes uniquely downregulated in GDM tissues are yet to be characterized but include a peroxide-responsive transcriptional repressor and a recombinase family protein, while the one uniquely upregulated DEG in controls is a sugar transporter (Fig. 2c–e, Supplementary Table 1). When comparing placental GBS vs. uterine GBS, we identified 50 DEGs of which 42 were shared between control and GDM groups and included upregulation of genes involved in metabolite (sugar, amino acids) transport and DNA replication and repair, with

downregulation of genes involved in transcriptional regulation and sugar transport (Fig. 2f–h, Supplementary Table 2). Three DEGs were uniquely regulated in GDM mice and five were uniquely regulated in control mice (Fig. 2f–h, Supplementary Table 2). The three genes uniquely downregulated in GDM tissues are yet to be characterized but include an LrgB family protein which has been shown to be repressed by high glucose conditions and modulates bacterial cell death and lysis in other streptococcal species[44]. Three of the five uniquely upregulated GBS DEGs in control placentae are uncharacterized, with two involved in transcriptional regulation or amino acid export (Fig. 2f–h, Supplementary Table 2). Uterine GBS and placental GBS shared 5 DEGs (SAK_RS07995, SAK_RS00885, SAK_RS00825, SAK_RS08150, SAK_RS00940) which include an acetyltransferase, L-lactate dehydrogenase, ribose transporter, YneF family protein, and a transcription factor respectively.

One upregulated gene in sequencing batch 1, SAK_RS10730 (here named *yfhO*), encodes a YfhO family protein which is predicted to have 14 transmembrane helical domains according to AlphaFold2[45,46] modeling (Fig. 3a) and has been identified as a GT-C fold glycosyltransferase by prediction program HHpred[47]. Members of this family catalyze the transfer of sugar moieties from lipid-linked sugar donors to diverse acceptors[48]. In *Bacillus subtilis* and

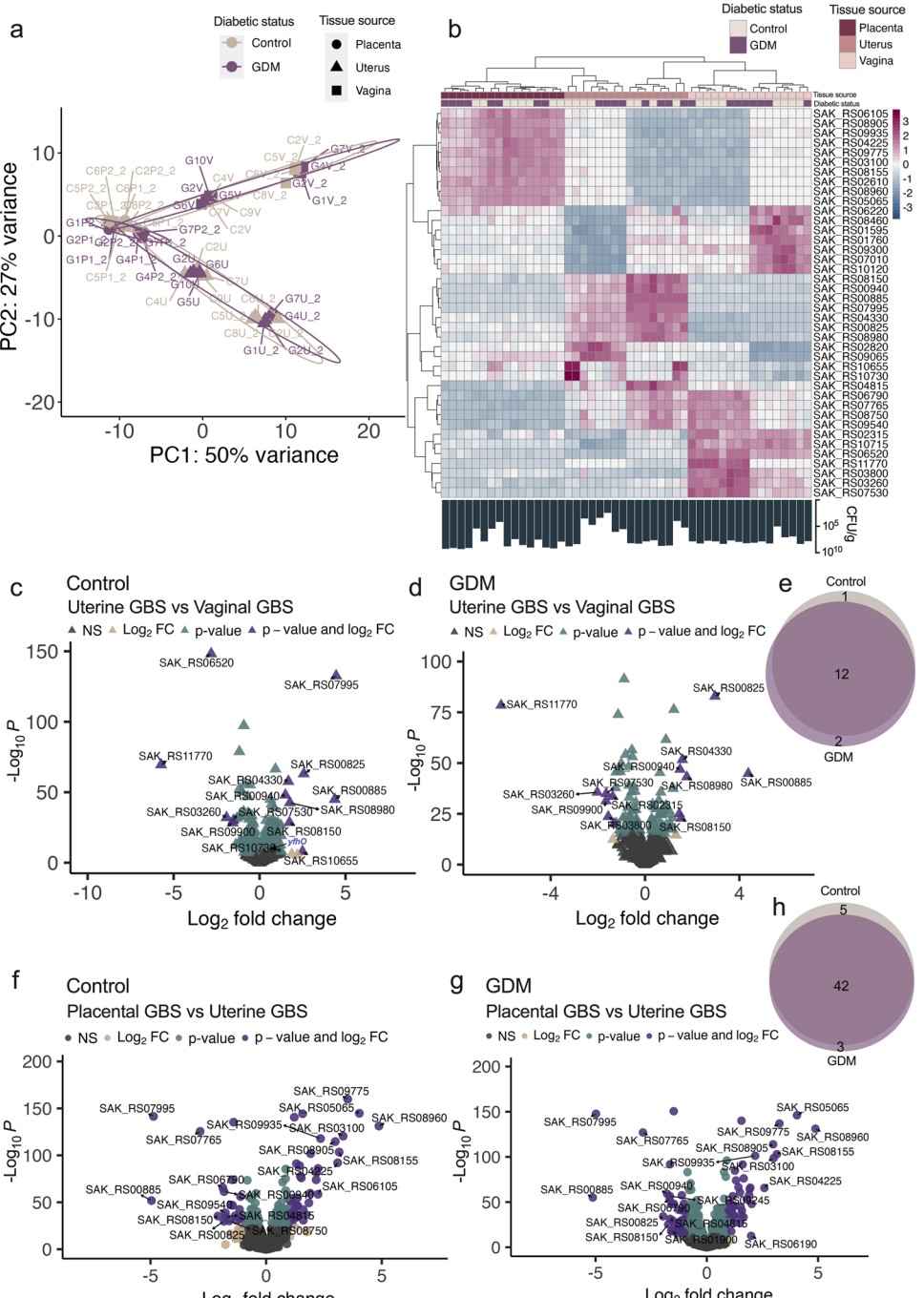

**Fig. 2 | GBS transcriptional profiling in control and gestational diabetic pregnancy identifies candidate genes important for ascension to the gravid uterus and placental invasion. a** Principal component analysis of GBS transcriptional profiles from vaginal (V), uterine (U), or placental tissue (P) of pregnant control (C) or GDM (G) mice on E17.5 (n = 8/group, two independent experiments for vaginal and uterine and one experiment for placental tissues). Each point represents a sample from an individual mouse, with vaginal-uterine-placental pairs indicated by numeric label. **b** Heatmap of 40 genes differentially expressed in GBS from uterine or placental tissue compared to GBS from vaginal tissue of pregnant controls and GDM mice. Tissue burden is indicated below the heatmap. Volcano plot of differentially expressed genes (DEGs) in uterine GBS vs. vaginal GBS in (**c**) pregnant controls or (**d**) GDM mice. **e** Venn diagram showing the proportion of GBS DEGs unique vs. shared between comparisons in (**c**, **d**). Volcano plot of differentially expressed genes (DEGs) in placental GBS vs. uterine GBS in (**f**) pregnant controls or (**g**) GDM mice. **h** Venn diagram showing unique vs. shared DEGs unique vs. shared between comparisons in (**f**, **g**). DEGs were identified via generalized linear model, Log₂ fold change >1 and Wald tests with FDR adjusted p value < 0.05. (**c, d, f, g**). DEGs were identified from independent analysis of experiment 1 and analysis of experiment 1 and 2 combined. Also see Supplementary Tables 1 and 2.

*Listeria monocytogenes*, YfhO homologs are involved in glycosylation of teichoic acid polymers that contribute to virulence of Gram-positive bacteria[49,50]. We assessed the role of YfhO in ascending uterine infection using an isogenic allelic exchange mutant expressing a kanamycin resistance cassette in place of *yfhO* (Δ*yfhO*).

Compared to parental A909, Δ*yfhO* showed decreased GBS burdens and rate of ascension to the uterus in control and GDM mice, without deficits in vaginal colonization or ascension to the cervix (Fig. 3b, c). Δ*yfhO* maintained the capacity to invade GDM placental tissues, but not control placentae (Fig. 3d), and was not detected in

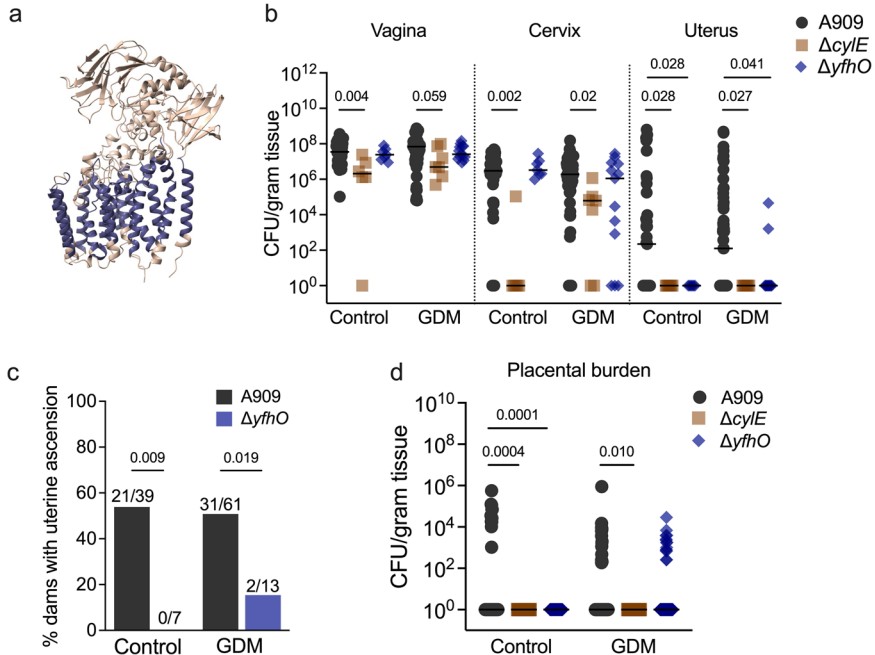

**Fig. 3 | Loss of YfhO limits GBS uterine ascension and placental invasion.**
**a** Predicted AlphaFold2 structural modeling of YfhO (SAK_RS10730) protein structure. **b** E17.5 GBS burdens in the reproductive tract of mice challenged with WT A909, Δ*cylE*, or Δ*yfhO*. **c** Proportion of dams with uterine ascension (detectable CFU) of WT A909 or Δ*yfhO*. **d** Placental GBS burden in control and GDM groups. Data for WT A909 are aggregated from all experiments (**b**–**d**), of which 5 pregnant controls and 10 GDM are from concurrent GBS mutant experiments (*n* = 39 pregnant controls, *n* = 61 GDM), mice challenged with Δ*cylE* (*n* = 6 pregnant controls, *n* = 7 GDM) are from 2 independent experiments, and mice challenged with Δ*yfhO*

(*n* = 7 pregnant controls, *n* = 13 GDM) are from 5 independent experiments. For placental burdens (d), resulting placentas from dams challenged with WT A909 (*n* = 63 pregnant controls, *n* = 116 GDM), Δ*cylE* (*n* = 44 pregnant controls, *n* = 54 GDM), and Δ*yfhO* (*n* = 58 pregnant controls, *n* = 103 GDM) were subjected to CFU quantification. Points represent individual samples and lines indicate medians (**b**, **d**). Source data are provided as a Source Data file. Data were analyzed by Kruskal-Wallis followed by Dunn's multiple comparisons test (**b**, **d**) and two-sided Fisher's exact test (**c**).

fetal livers from either group (0/60 control, 0/107 GDM fetal livers). To test whether GDM mice were more broadly susceptible to GBS colonization and fetal dissemination, we assessed the role of the well-characterized β-hemolysin/cytolysin using an isogenic mutant of *cylE*, a gene required for β-H/C production[51]. A909 Δ*cylE*, previously shown to have attenuated fetal dissemination[12] and vaginal colonization in non-pregnant mice[11], was equally attenuated in control and GDM mice (Fig. 3b, d).

**Gestational diabetes skews host transcription and cytokine profiles**
Host transcriptional profiles also clustered in a tissue-specific manner independent of diabetic status (Fig. 4a). Nevertheless, GDM mediated changes to vaginal, uterine and placental transcriptomes (Fig. 4b–d). When comparing the vaginal transcriptome of GDM mice to controls, we identified nine DEGs of which 7 were significantly downregulated in the diabetic group including *Chil4*, a positive regulator of chemokine production and Class Ib MHC antigen *H2-Q6*, associated with antigen processing and T cell-mediated cytotoxicity (Fig. 4b, Supplementary Table 3). Significantly upregulated pathways involved IFNγ responses, DNA repair, apoptosis, and hypoxia signaling suggesting greater GBS-associated vaginal tissue damage and inflammation in GDM hosts (Fig. 4e). GDM uteri had 14 DEGs compared to control uteri and included downregulated *RLN1* encoding Relaxin 1, a hormone that inhibits uterine contractions[52] (Fig. 4c, Supplementary Table 3). Significantly upregulated pathways in the GDM uterus included immune responses (IFNγ and TNFα signaling) and significantly downregulated pathways involved metabolic and protein homeostatic responses (Fig. 4f). Three significantly upregulated genes were detected in GDM placentae (Fig. 4d) which are involved in proteolysis, post-translational modification and metabolic processes. Significantly upregulated

pathways in GDM placentae included several immune responses (IL-2, TNFα, IFNγ, IL-6, TGF-β) and downregulation of glycolysis and cell cycle pathways (Fig. 4g).

Together, transcriptional profiles suggested altered GBS-associated immune responses in GDM hosts. Indeed, a multiplex cytokine assay showed that GDM mice had dysregulated vaginal, uterine and placental immune responses to GBS. GDM mice had significantly greater vaginal levels of G-CSF and KC and lower IL-2, and significantly less uterine KC compared to controls (Fig. 5a, Supplementary Fig. 3). Compared to respective mock-infected dams, GBS inoculation elevated vaginal KC, IL-1β, and IL-12p70 in both GDM and control dams. GDM mice had significant induction of IL-1α and G-CSF, whereas controls significantly induced IL-2, compared to mock-infected counterparts. In contrast, GBS led to significant decreases in several uterine cytokines, of which IL-9, IFNγ, and TNFα, were diminished in GDM and controls compared to respective mock-infected groups (Fig. 5a, b). Uterine IL-1α was uniquely induced in GBS-infected GDM dams compared to mock-infected GDM dams. GBS induced a robust cytokine response in placentae, with a greater amount of significantly increased cytokines observed in the GDM group (Fig. 5a). To determine cytokine responses associated with limiting GBS infection, we disaggregated the data into GBS positive and GBS negative tissues and compared cytokine levels (Fig. 5c–e, Supplementary Fig. 4). GBS positive placentae in control dams had overwhelmingly greater levels of 14 cytokines relative to GBS negative uteri. In contrast, only IL-2 was significantly greater in GBS positive placentae in GDM dams compared to GBS negative placentae (Fig. 5c–e). Cumulatively, these findings reveal tissue-specific cytokine responses to GBS and show that GDM dysregulates this arm of reproductive immunity.

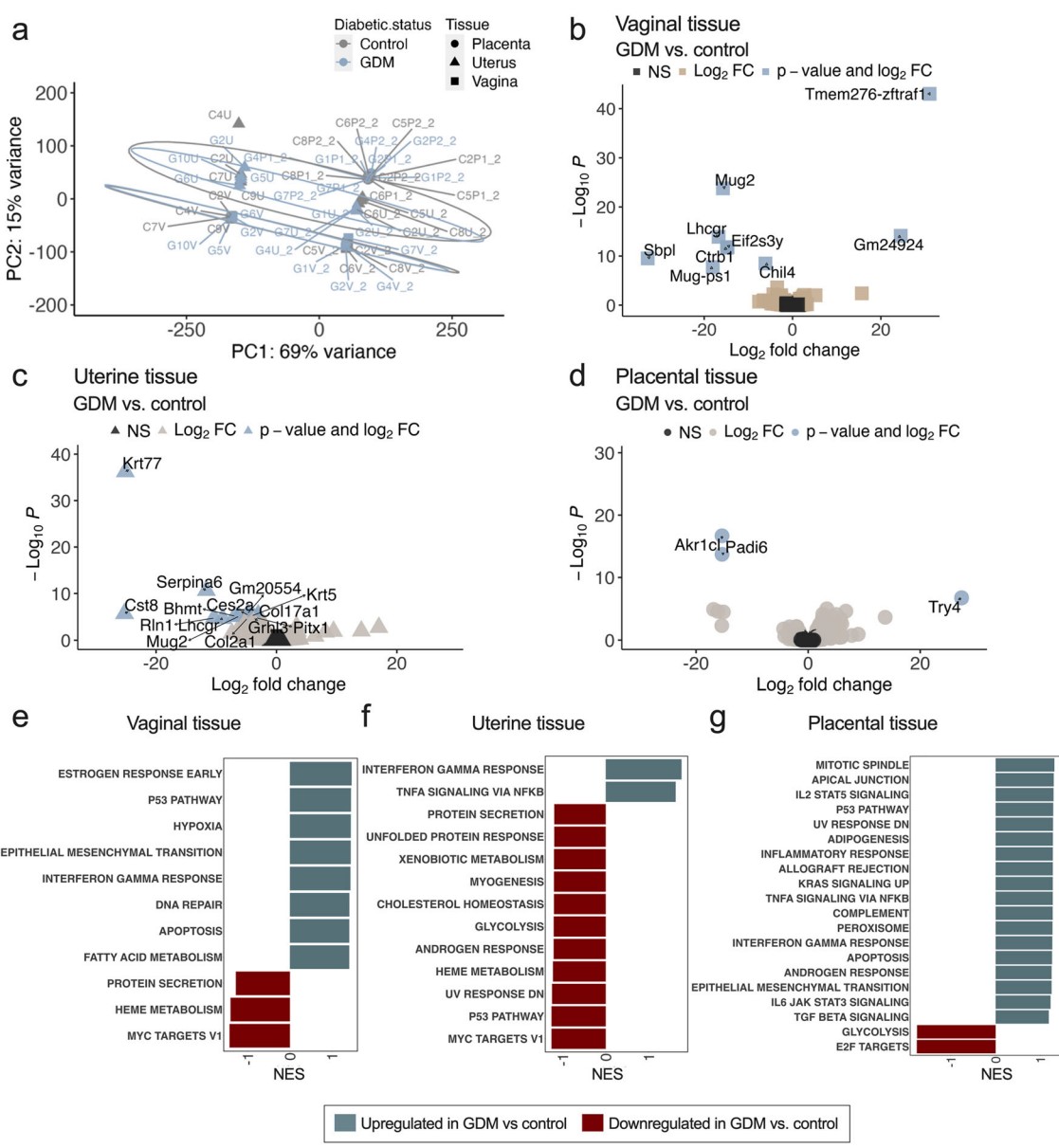

**Fig. 4 | The reproductive transcriptional landscape is altered in gestational diabetic mice during GBS challenge. a** Principal component analysis of host transcriptional profiles from vaginal (V), uterine tissue (U), or placental tissue (P) of pregnant control (C) or GDM (G) mice on E17.5 ($n = 8$/group, two independent experiments for vaginal and uterine tissues, one independent experiment for placental tissues). Each point represents a tissue from an individual mouse, with vaginal-uterine-placental pairs indicated by numeric label. Volcano plot of differentially expressed genes (DEGs) in (**b**) vaginal tissue, (**c**) uterine tissue, and (**d**) placental tissue from GDM mice vs. pregnant controls. Gene set enrichment analysis of (**e**) vaginal, (**f**) uterine, and (**g**) placental tissues in gestational diabetic mice vs. pregnant controls. NES = Normalized Enrichment Score. DEGs were identified via generalized linear model, Log₂ fold change >1 and Wald tests with FDR adjusted $p$ value < 0.05. fGSEA was performed with a gene set minimum of 15, a gene set maximum of 500, 10,000 permutations, and the Hallmark gene set collections from the Molecular Signatures Database.

## Gestational diabetes alters immune cell profiles upon GBS challenge

Next, we evaluated immune cell infiltration in E17.5 vaginal, uterine and placental tissues by flow cytometry using a 19-marker antibody cocktail to distinguish lymphoid and myeloid lineage subsets (Fig. 6a). There were no differences in the composite CD45+ fraction between GDM and control samples across tissue types (Fig. 6b). Additionally, no differences between GDM and control maternal splenic immune populations were detected (Supplementary Fig. 5a). GDM mice displayed several significantly altered immune cell subsets proportions compared to controls including increased B cells in vaginal tissues (Fig. 6c), and decreased NK and regulatory T cells in uterine tissues (Fig. 6d), with no differences in total immune cell counts in either tissue (Supplementary Fig. 5b–e). Uterine NK (uNKs) cells, defined as uterine NK1.1+ CD69+ cells[53], were enriched in uteri of both groups (Fig. 6e), but GDM mice displayed a significantly decreased proportion of uNKs positive for activation marker CD25 (Fig. 6f). While proportions of placental immune cells were similar between groups, total cell counts revealed that GDM placentae had significantly fewer CD45+ cells, and several immune subpopulations including neutrophils and NKs (Fig. 6g–h, Supplementary Fig. 5b, c, f). Next, we probed the involvement of neutrophils and NKs in ascending fetal infection. Neutrophils or NK cells were depleted with α-Ly6G or α-NK1.1 antibodies respectively with one dose given before and an additional dose given during GBS challenge (Fig. 7a). Specific local depletion of NK cells (Fig. 7b, c) or neutrophils (Fig. 7b, d), was confirmed by flow

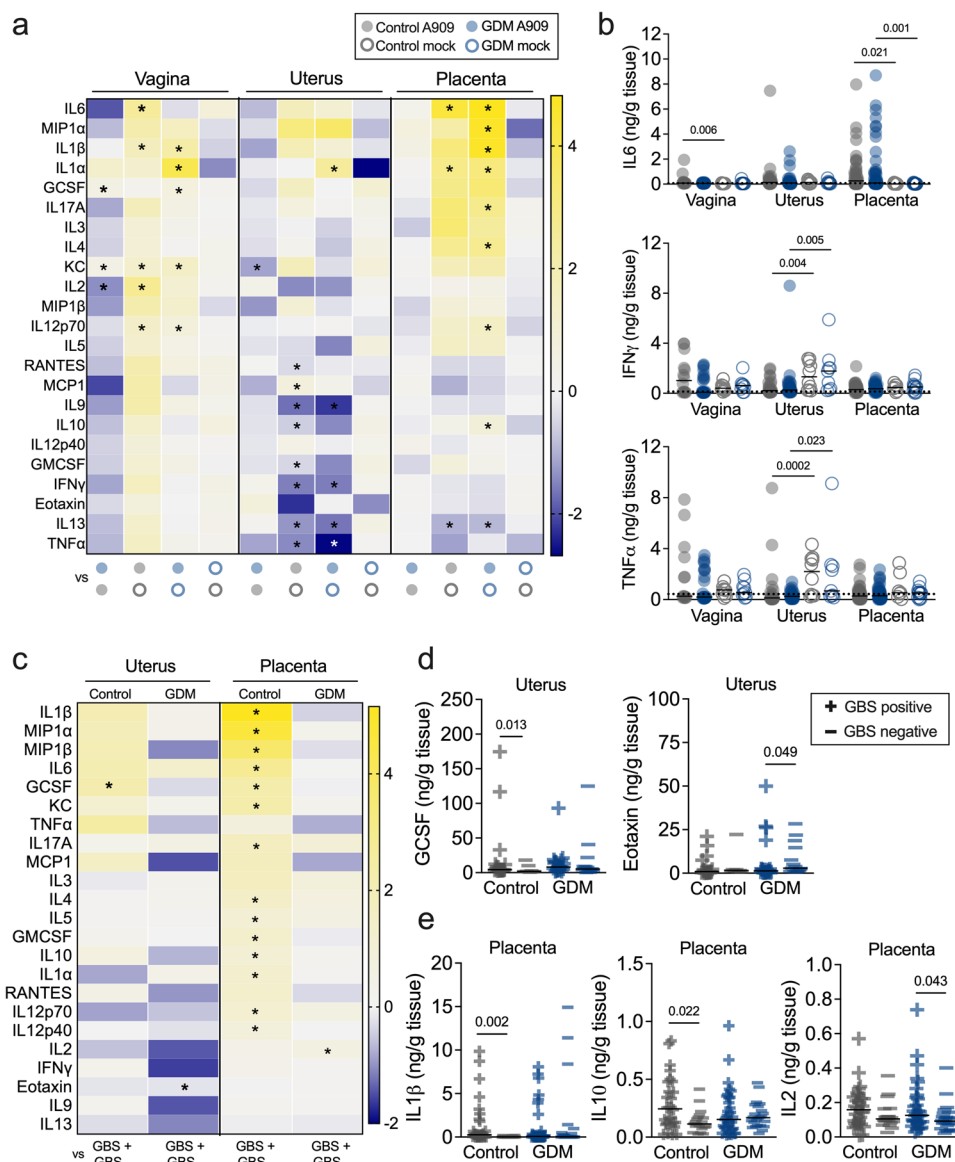

**Fig. 5 | Cytokine responses are altered in gestational diabetic mice in response to GBS challenge. a** Heatmap of 23 cytokines in vaginal, uterine and placental tissues on E17.5 from pregnant controls and GDM mice that were inoculated with A909 or mock-infected. Scale bar reflects Log 2-fold change. **b** Select cytokines that were significantly different between groups. Other cytokines are shown in Supplementary Fig. 3. **c** Heatmap comparing tissue cytokine levels by GBS status on E17.5. Scale bar reflects Log 2-fold change. Select cytokines that were significantly different in (**d**) uterine tissue and (**e**) placentae based on whether GBS CFU were detected or not. Other cytokines are shown in Supplementary Fig. 4. Data (**a**, **b**) are from 5 independent experiments with *n* = 14 vaginal, 27 uterine, and 48 placental tissue samples from infected controls, *n* = 25 vaginal, 37 uterine and 57 placental tissue samples from infected GDM, and *n* = 10 vaginal, 10 uterine and 7 placental tissues from mock-infected controls and 9 vaginal, 9 uterine, and 13 placental tissues from mock-infected GDM mice. Points represent individual samples and lines indicate medians (**b**, **d**, **e**). Source data are provided as a Source Data file. Cytokine data were analyzed by Kruskal-Wallis test followed by a two-stage linear step-up procedure of Benjamini, Krieger and Yekutieli to correct for multiple comparisons by controlling the false discovery rate (<0.05) (**a**, **b**), or by two-tailed Mann-Whitney *t* test (**c**–**e**). For heatmaps (**a**, **c**), \**p* < 0.05. Exact *p* values are provided in Supplementary Figs. 3 and 4.

cytometry of uterine tissues (Supplementary Fig. 5g). NK depletion did not impact maternal tissue burdens in GDM or controls (Fig. 7e–g). Neutrophil depletion led to greater GBS load in vaginal tissues of controls (Fig. 7e), with no effect on cervical or uterine burdens in both groups (Fig. 7f, g). In GDM mice, NK depletion had protective effects against fetal infection shown by significantly fewer GBS-positive feto-placental tissues (Fig. 7h–j) and decreased bacterial load in fetal livers (Fig. 7k). Controls displayed reduced GBS invasion of fetoplacental tissues overall compared to the GDM group (as in Fig. 1g, h) with NK depletion significantly reducing the incidence of fetal liver GBS invasion (Fig. 7j). In contrast, neutrophil depletion was inconsequential for fetal dissemination in the GDM group, but in controls, led to increased

frequency of placental GBS positivity (Fig. 7h–k). Additionally, preterm birth (delivery of pups at or before E17.5) was observed in 2/5 GDM dams treated with anti-Ly6G but this was not observed in the corresponding controls (0/7 dams). Collectively, these findings suggest distinct roles for NKs and neutrophils in controlling GBS infection at the maternal-fetal interface in healthy pregnancy and in susceptible hosts.

## Gestational diabetes augments adverse neonatal outcomes
Previously described mouse models of intra-vaginal GBS inoculation and subsequent transmission to vaginally-delivered neonates have shown adverse outcomes such as decreased survival, stunted weight,

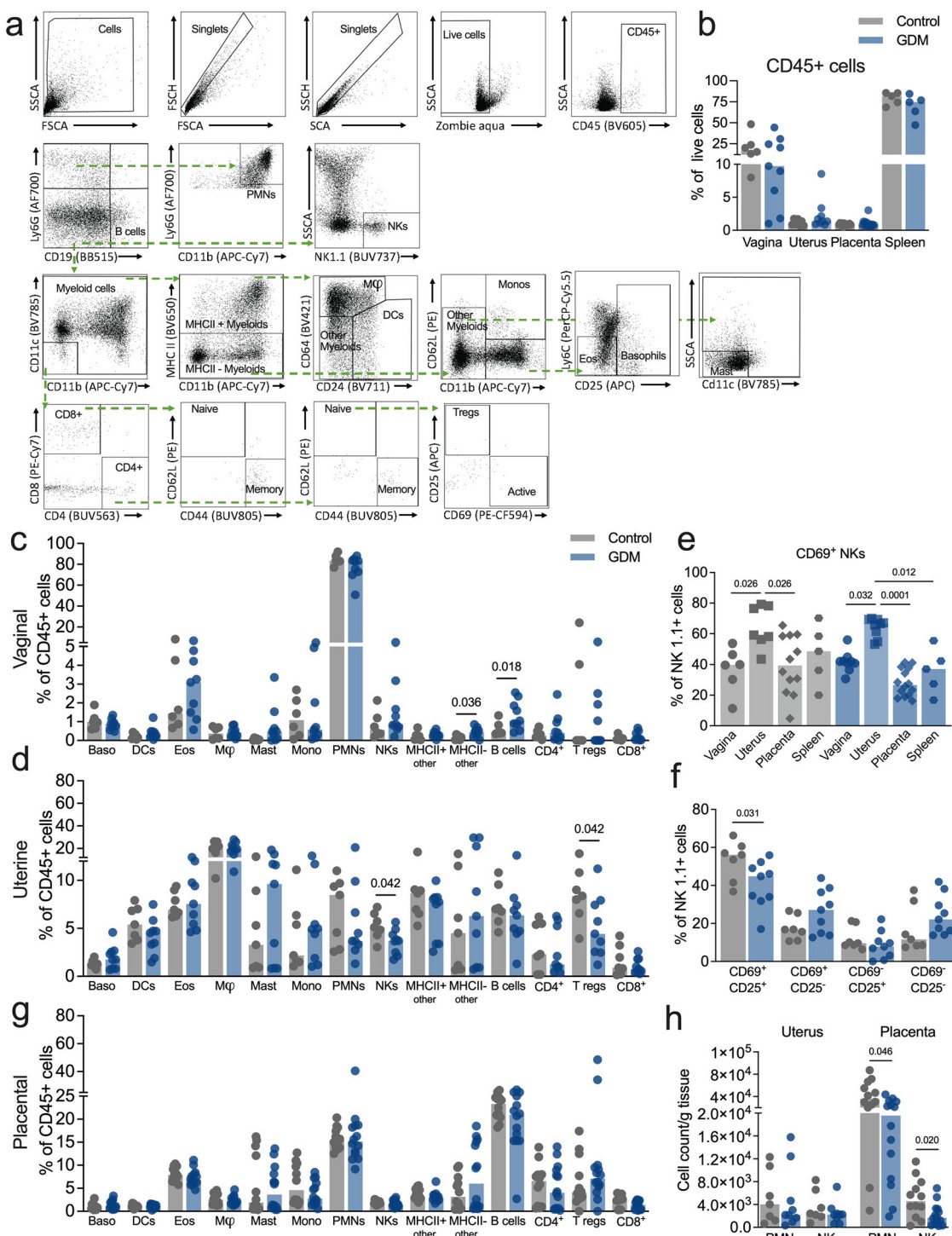

**Fig. 6 | Immune cell recruitment is dysregulated in gestational diabetic mice.** **a** Gating strategy for assessing recruitment of basophils (Baso), dendritic cells (DCs), eosinophils, macrophages (Mφ), mast cells, monocytes, neutrophils (PMNs), NK cells (NKs), B cells, CD4⁺ T cells, CD4⁺ regulatory T cells (T regs) and CD8⁺ T cells by flow cytometry. **b** Frequency of CD45+ cells. Immune cell frequencies in (**c**) vaginal and (**d**) uterine tissues in GBS-infected dams. **e** Frequencies of CD69 + NK cells across tissues. **f** Uterine NK cell proportions stratified by CD69 and CD25 expression. **g** Immune cell frequencies in placentae collected from GBS-infected dams. **h** Total cell counts of neutrophil and NK cells from placentae. Data (**b–h**) are from 3 independent experiments with each point representing an individual mouse sample (*n* = 7 pregnant controls and 9 GDM, with 1–2 placentae per dam for a total of *n* = 12 control placentae and 14 GDM placentae). Source data are provided as a Source Data file. Data were analyzed by two-tailed Mann–Whitney *t* tests per tissue (**b–d**, **f–h**) or Kruskal-Wallis followed by Dunn's multiple comparisons test (**e**).

and systemic infection[54,55]. Male pups have increased circulating IL-1β and placental IL-1β and CXCL1 in rat models of *in utero* GBS exposure[56], but this was not replicated in our study (Supplementary Fig. 5h, i). To examine the effect of GDM on offspring outcomes, we monitored neonatal outcomes born to GBS-inoculated dams through postnatal

day 7. Neonates born to GDM dams had significantly worse survival and stunted weight in the first week of life compared to neonates from control dams (Fig. 8a, b). To track GBS transmission and systemic disease, we measured GBS burden in neonatal intestines (indicating colonization) and livers (indicating invasive disease) upon death or on

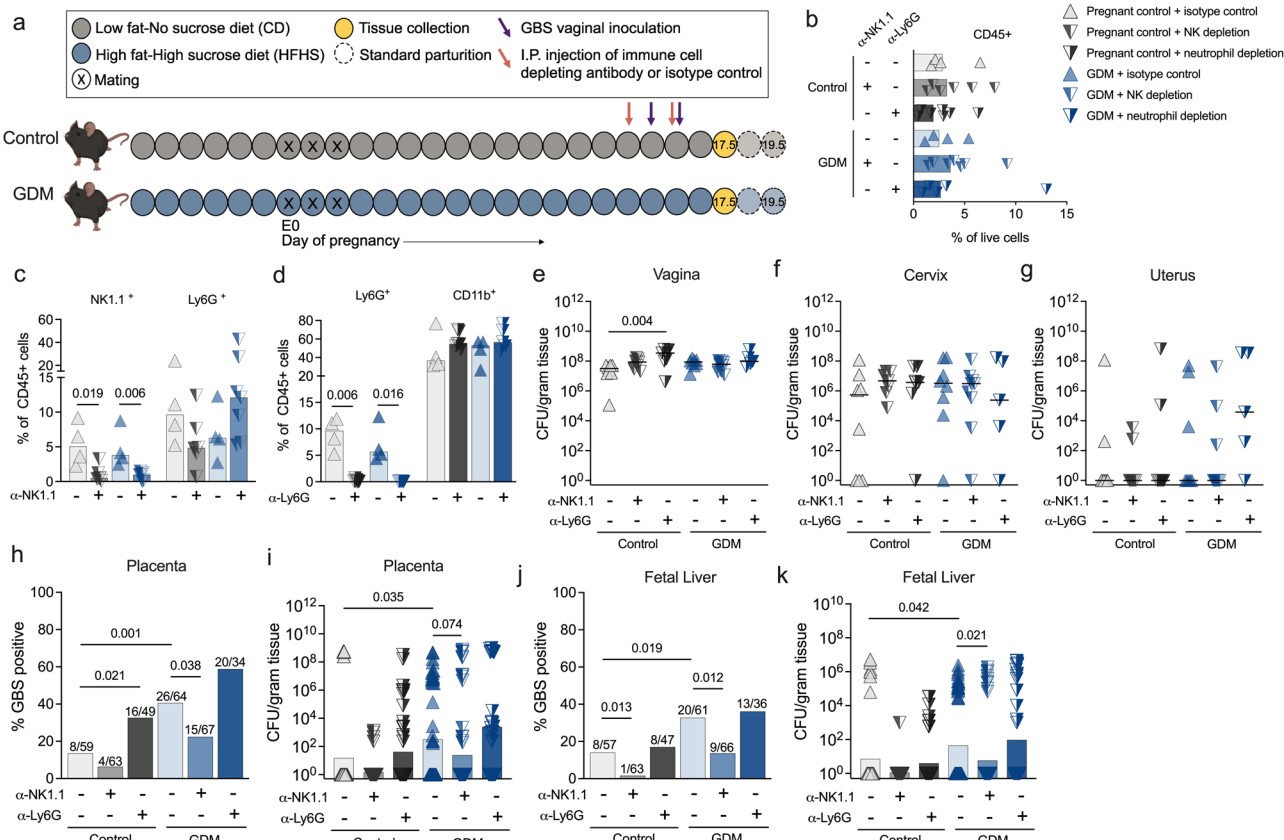

**Fig. 7 | NK and neutrophil depletion differentially impact ascending GBS infection. a** Schematic of experimental timeline for immune cell depletion and subsequent GBS challenge in GDM and pregnant control (ctrl) mice. Mouse image created with BioRender.com. **b** Frequencies of uterine CD45+ cells in NK-depleted (α-NK1.1), neutrophil-depleted (α-Ly6G), and isotype-treated mice. **c** Frequencies of uterine NK1.1+ and Ly6G+ cells in isotype-treated or NK-depleted mice. **d** Frequencies of uterine Ly6G+ and CD11b+ cells in isotype-treated or neutrophil-depleted mice. **e–g** GBS burdens of maternal reproductive tract tissues with (α-NK1.1 or α-Ly6G) or without (isotype control) immune cell depletion in GDM and pregnant control mice. **h** GBS positive proportions (detectable CFU) and (**i**) burdens of placentae with or without immune cell depletion in GDM and pregnant control mice. **j** GBS positive proportions (detectable CFU) and (**k**) burdens of fetal

livers with or without immune cell depletion in GDM and pregnant control mice. Data for isotype treated mice are pooled from 7 independent experiments; 4 NK depletion experiments and 3 neutrophil depletion experiments. For NK depletion experiments, n = 7 pregnant controls and 9 GDM mice that were NK-depleted, n = 7 pregnant controls and 6 GDM mice that were isotype-treated. For neutrophil depletion experiments, n = 7 pregnant controls and 5 GDM mice that were neutrophil-depleted, n = 7 pregnant controls and 6 GDM mice that were isotype-treated. Experimental numbers for placenta-fetal pairs in each group are given as denominators in (**h, j**). Source data are provided as a Source Data file. Data were analyzed by multiple two-tailed Mann–Whitney t tests with correction for multiple comparisons (**c, d**), by Kruskal-Wallis test with a post-hoc Dunn's multiple comparisons test (**e–g, i, k**), or by two-sided Fisher's exact test (**h, j**).

postnatal d7 in surviving pups. We found no differences in intestinal GBS colonization or systemic infection between GDM and control neonates regardless of time of collection (Fig. 8c, d). There were also no differences in hours to delivery following first GBS inoculation (Fig. 8e). A stark disparity in female pup survivorship likely contributed to observed differences between GDM and control groups (Fig. 8f). While there were no sex-dependent differences in d7 pup weights (Fig. 8g), GDM males had significantly greater liver burden on d7 compared to GDM females, with no sex-effects observed in controls (Fig. 8h). Together, these data suggest that gestational diabetic pregnancies exhibit increased susceptibility of GBS dissemination to the fetal compartment with subsequent worse neonatal survival and growth.

### Vaginal microbiota dynamics and taxa linked with GDM and GBS invasion

The vaginal microbiota composition is associated with pregnancy complications and birth outcomes across patient demographics and geographic regions[57–59] and several clinical studies have demonstrated that women with GDM exhibit distinct distribution of vaginal taxa compared to healthy controls[37,38,60,61]. By collecting vaginal swabs every

3 days from diet introduction through E14 of pregnancy of mice from the pup outcomes and E17.5 experiments above, we longitudinally characterized the vaginal microbiota in this murine model with two primary objectives: (A) to define how the murine vaginal microbiota changes throughout non-diabetic and diabetic pregnancy and (B) to determine if there are signatures of vaginal communities that are protective or permissive against GBS *in utero* dissemination. Consistent with reports in nonpregnant mice[62,63] the predominant taxa in pregestational C57BL/6J mice included *Staphylococcus*, *Lactobacillus*, *Enterococcus*, *Corynebacterium*, and *Streptococcus* spp. (Fig. 9a, b, Supplementary Fig. 6a, b). We also detected *Enterobacteriaceae*, *Bacillus* and *Lactococcus* spp. (Fig. 9a,b). In controls, the vaginal microbiota increased in alpha diversity between the day prior to mating (d-1) and E14, whereas no change occurred in the GDM group (Fig. 9c). When we compared all premating timepoints (d-7 to d-1) to gestational or post-mating timepoints (d2-d14), increased alpha diversity was specific to pregnant control mice, and decreased biomass (total mapped reads) was observed in pregnant control and GDM mice but absent in the non-pregnant cohort (Fig. 9d, e, Supplementary Fig. 6c−f). This indicates a pregnancy-specific phenomena and confirms GDM-mediated disruption of diversity fluctuation of the

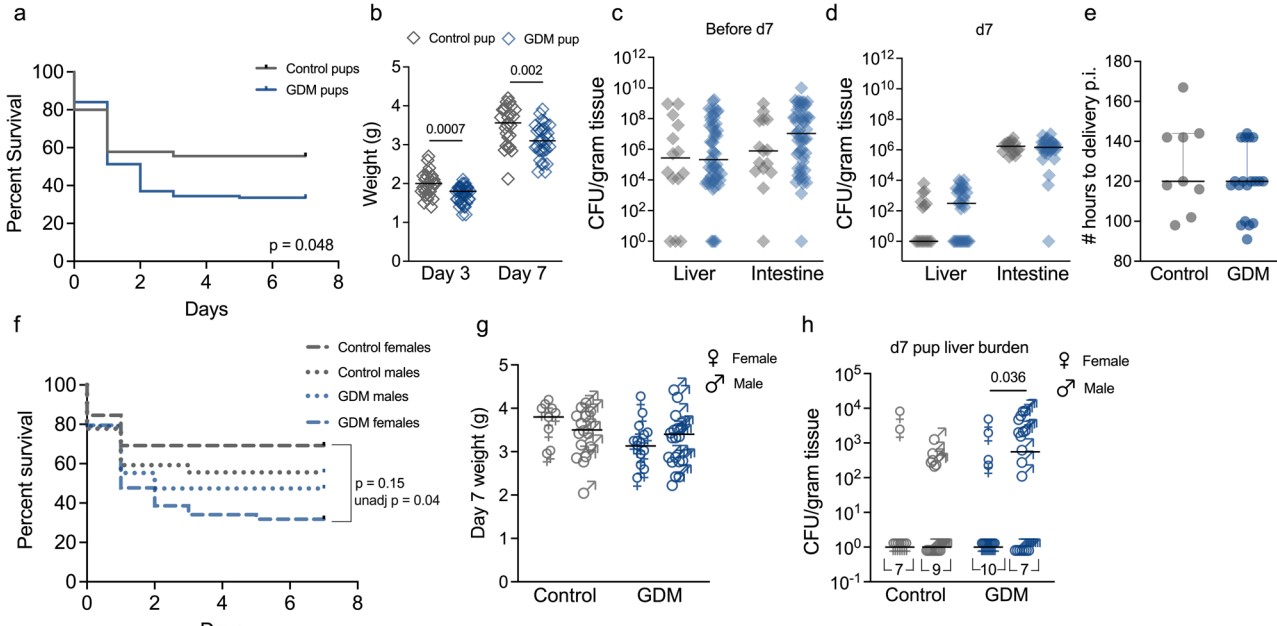

**Fig. 8 | Gestational diabetes worsens neonatal outcomes associated with GBS infection. a** Survival of neonates from GDM or control dams that were inoculated with GBS on E14.5 and E15.5. *n* = 45 offspring from 9 control dams and 119 offspring from 19 GDM dams. All remaining neonates were sacrificed on d7 for burden quantification. **b** Neonatal weights on days 3 and 7 of life, *n* = 25 control pups and 35−42 GDM pups. GBS burden in pup livers and intestines (**c**) near time of death before d7 (*n* = 15 control neonates and 54 GDM neonates), or (**d**) on d7 (*n* = 17 control neonates and 35 GDM neonates). **e** Time to delivery following initial GBS inoculation. **f** Pup survival, (**g**) d7 pup weights, and (**h**) pup GBS liver burdens stratified by sex. All data are from 4 independent experiments. Points represent individual samples and lines indicate medians. Source data are provided as a Source Data file. Data were analyzed via Kaplan-Meier survival analysis with Holm-Šídák correction for multiple comparisons (**a**, **f**) and two-sided Mann–Whitney *t* test (**b−e**, **g**, **h**).

pregnant murine vaginal microbiota (Fig. 9d, e). Additionally, greater alpha diversity was associated with protection against GBS perinatal invasive disease in both GDM and control dams (Fig. 9f). There were no GDM-mediated differences in intra-mouse pairwise Bray-Curtis (BC) dissimilarity between consecutive timepoints (Fig. 9g); however, clustering by BC dissimilarity indices at pre-gestation, early gestation and mid-gestation revealed several key findings: (A) that differences in vaginal communities were driven by 4 taxa (*Staphylococcus*, *Enterococcus*, *Lactobacillus*, and *Bacillus*), (B) diabetic status did not impact overall vaginal community structure, and (C) vaginal communities drifted away from *Staphylococcus succinus* dominance towards later points in gestation (E11, E14) in both non-diabetic and diabetic mice (Fig. 9h−k, Supplementary Fig. 6g−j). *S. succinus* relative abundance significantly decreased by mid-gestation for all pregnant mice irrespective of diabetic status (Fig. 9k), but no changes occurred in non-pregnant mice (Supplementary Fig. 6j). Analysis of Composition of Microbes (ANCOM) revealed *Enterococcus* as a feature that was specific to GDM pregnancy (Fig. 9l, Supplementary Fig. 6k, l). Additionally, ANCOM of samples from dams that had fetal GBS invasion vs. those that did not revealed that vaginal *Lactobacillus* and *Enterobacteriaceae* varied significantly between groups (Fig. 9m). Control dams that had perinatal GBS invasion of offspring had significantly less *Lactobacillus* at mid-gestation (Fig. 9n). Increased *Enterobacteriaceae* during early gestation was associated with perinatal invasive disease in both controls and GDM groups (Supplementary Fig. 6m). The GDM cohort had significantly greater relative abundance of *Enterococcus* on E2 suggesting an early enterococcal bloom (Fig. 9o, Supplementary Fig. 6n). By E17.5, a significantly greater proportion of GDM dams (44% vs. 12.5% of controls) experienced spontaneous uterine-fetal co-ascension by *Enterococcus*, detected by blue colony growth on differential medium, in dams inoculated with A909 but not CNCTC 10/84 (Fig. 9p, Supplementary Fig. 2c). Together, these analyses reveal that

GDM uniquely alters vaginal microbiota dynamics in pregnancy and implicates specific taxa in influencing GBS perinatal disease.

## Discussion

Clinical data provide strong associations between diabetes in pregnancy and GBS vaginal carriage[24,25] and neonatal disease[26], but the molecular mechanisms driving these associations are undefined. Here, we describe the first animal model interrogating group B *Streptococcus* pathogenesis in gestational diabetes that mirrors the natural course of GBS vaginal colonization, *in utero* dissemination, and neonatal transmission. We observed that GDM mice had a significantly greater risk of GBS fetal infection and worse neonatal outcomes. We identified dysregulated maternal immunity, an altered maternal vaginal microbiota, and discordant GBS transcriptional adaptation including a uncharacterized GBS gene, *yfhO*, as mechanistic contributors to GBS disease in GDM hosts. Prior reports of GBS vaginal carriage in diabetic pregnant women (gestational and/or pregestational diabetes) had mixed findings with some suggesting an increased risk[24,25], while others found no association[64,65]. These conflicting findings are possibly explained by GDM heterogeneity including varied severity of hyperglycemia and insulin dependency, immune dysfunction, GBS strain-dependent behavior, and other compounding biological and socioeconomic differences in study cohorts. In a recent meta-analysis of 19 studies, we found that GDM significantly increases maternal risk of rectovaginal GBS carriage[66]. Although this was not recapitulated in our model of genetically and phenotypically similar GDM hosts[40,41], this model permitted mechanistic resolution of GBS reproductive tract ascension and perinatal disease. Observed ranges in maternal and fetal burdens are likely related to the 2−3 day mating window used to maximize successful mating. Timing and dosage of GBS impact murine birth outcomes with earlier (E13, E14) and greater inocula associated with preterm birth[12,15] compared to later (E17, E18) and lower inocula[54].

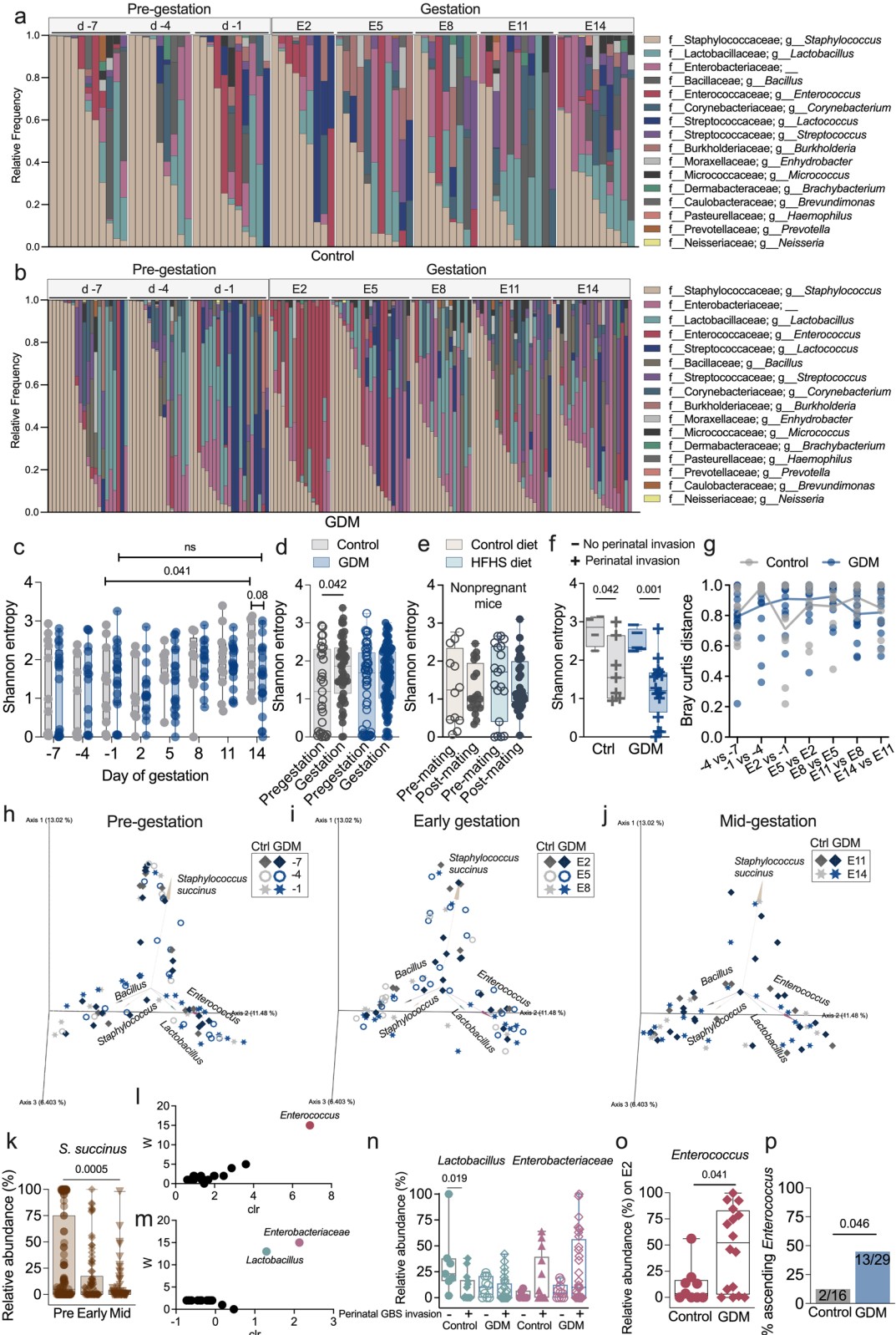

Despite the range in gestational age and two inoculations, we only observed 2 cases of preterm birth and both cases were in neutrophil-depleted GDM dams. Ultimately, the range in gestational age and tissue burdens provide a clinically relevant model reflective of GBS exposure patterns experienced by women.

Our data suggest that poor GDM pup outcomes (reduced survival and weight gain) were not driven by differences in postnatal GBS

burdens, but perhaps were driven by an earlier disadvantage *in utero* shown by increased prenatal bacterial load. Our findings agree with a clinical report showing a 2-fold increased risk for early onset GBS sepsis or pneumonia for infants born to GDM mothers[26]. In humans, neonatal sex impacts GBS outcomes; male neonates with invasive GBS have a greater long-term risk of neurodevelopmental impairment and/ or epilepsy[67,68]. We did not assess neonatal neurological development,

**Fig. 9 | Pregnancy and gestational diabetes influence the vaginal microbiota composition and specific taxa correlate with GBS offspring dissemination.**
**a** Control (Ctrl) and (**b**) GDM mice were swabbed every three days from before mating (d-7) until E14 and relative frequency of taxa, collapsed to the genus level, are shown. **c** Shannon entropy of vaginal communities. **d** Composite Shannon entropy of all timepoints before (d-7 to −1) and during gestation (E2-E14).
**e** Composite Shannon entropy of all timepoints before mating vs. after mating in non-pregnant mice on matched diets. **f** Shannon entropy on E14 of mice with or without GBS dissemination to offspring (GBS detected in fetal or pup livers). **g** Bray Curtis distances between sequential timepoints per mouse. Bray Curtis distance matrix principle coordinate analysis plots at (**h**) pre-gestation (d-7 to d-1), (**i**) early gestation (E2-E8), and (**j**) mid-gestation (E11-E14). **k** Relative abundance of *S. succinus* with GDM and control samples combined. **l** Analysis of Composition of Microbes (ANCOM) based on GDM status. **m** ANCOM based on GBS offspring dissemination. **n** Relative abundances of *Lactobacillus* and *Enterobacteriaceae* at mid-gestation, stratified by GBS dissemination. **o** Relative abundance of

*Enterococcus* on E2. **p** Proportion of dams with *Enterococcus* (detectable CFU) in uterine or fetal tissues. Points represent individual samples and lines indicate medians. Box-and-whisker plots extend from 25th to 75th percentiles and show all points (**c–f**, **k**, **n**, **o**). Data are from 4 independent experiments, $n = 11$ controls, $n = 21$ GDM mice, $n = 8$ non-pregnant mice on control diet, $n = 11$ non-pregnant mice on HFHS diet. For GBS dissemination (**f**, **n**), $n = 7$ controls and 16 GDM with GBS, and $n = 4$ controls and 5 GDM without GBS. For endogenous *Enterococcus* ascension (**p**), $n = 16$ control and 29 GDM dams aggregated from experiments shown in Fig. 1. Source data are provided as a Source Data file. Unless otherwise noted, vaginal samples totaled 360 from the following groups; $n = 85$ control, 160 GDM, 34 non-pregnant mice on control diet, 53 nonpregnant mice on HFHS diet, 28 negative controls/blanks. Data were analyzed by two-tailed Welch's *t* test (**c**), two-tailed Mann−Whitney *t* test (**d–f**, **n**, **o**), two-sided Friedman's tests with correction for multiple comparisons (**g**), Kruskal-Wallis test with a post-hoc Dunn's multiple comparisons test (**k**) and two-sided Fisher's exact test (**p**).

but a sex dichotomy was detected at postnatal d7 with GDM males bearing higher invasive GBS burdens. It is possible that female pups with greater GBS invasive disease succumbed to GBS prior to d7 (Fig. 8f) and thus were underrepresented in the d7 liver samples; however, this speculation cannot be confirmed by the current experimental design. We also assessed sex-specific differences *in utero*. Females in the control group displayed greater GBS invasion, with no sex differences in GDM fetuses. Intrauterine position effects might shape these findings; females positioned near males have an altered hormonal environment that bears long-term effects on anatomy and physiology[69]. Key aspects of our future work are delineating the contributions of fetal and neonatal immunity to perinatal outcomes, which we suspect are negatively impacted by GDM. Another critical future direction is exploring the neonatal microbiota in this model, as there is evidence of GDM-mediated dysbiosis in human neonates including increases colonization by *Streptococcaceae* and *Enterococcaceae*[37,70,71], which may contribute to discrepant outcomes.

Previous studies have described GBS transcriptional adaptations to the murine non-gravid vaginal environment compared to in vitro culture conditions[17,23]. To our knowledge, this study provides the first comparison of GBS transcriptional responses between different tissues within the gravid reproductive tract. We identified 60 DEGs between vaginal, uterine, and placental sites. A gene associated with ribose metabolism (SAK_RS00825) was upregulated in vaginal tissue in our study in line with a previous study comparing GBS transcriptional profiles in vaginal tissues versus chemically defined medium[17]. Several DEGs overlapped with genes identified as critical for GBS colonization of the non-gravid murine reproductive tract using a GBS transposon mutant library including an acetyltransferase (SAK_RS07995), biotin synthase, a CHAP domain-containing protein (SAK_RS08980), amino acid ABC transporter (SAK_RS04225), LrgB family protein (SAK_RS06190), and the peroxide-responsive transcriptional repressor *perR*[23]. We also detected significantly higher transcription of *mrvR* in vaginal tissues from pregnant controls in batch 1 but not in combined analyses of both batches (Table S1); *mrvR* was recently reported to mediate nucleotide metabolism and provide GBS a growth advantage in amniotic fluid[21]. In high glucose media, GBS alters expression of genes involved in sugar transport, amino acid metabolism and transcription regulation[19]. In agreement, tissue-specific and group-specific transcriptional signatures of carbohydrate transport, metabolism, and transcription regulation in our study suggest metabolically unique environments within the reproductive tract. We did not detect higher tissue glucose in GDM mice on E17.5, three days post GBS challenge; However, it is possible that tissue glucose levels are different at other timepoints. Nevertheless, the upregulation of the GRP family sugar transporter (SAK_RS10655) and *yfhO*, a GT-C fold glycosyltransferase, in control uterine tissues but not GDM suggests relative sugar restriction in control tissues. YfhO has been linked to extracellular

stress responses in other bacterial species[72]. In this model, we found that *yfhO* was crucial for fetoplacental infection, with GDM conditions being permissive to this otherwise attenuated mutant. The 5 genes discordantly regulated in GDM uterine and placental tissues are yet to be characterized. Altered GBS-host interactions and increased pathogenicity in GDM is further supported by the upregulation of host cell death, cytokine signaling, and inflammatory pathways in GDM vaginal, uterine and placental tissues.

Multiple clinical studies have identified aberrant systemic immune profiles in women with GDM including increased peripheral neutrophils, B cells, and CD8+ T cells[29,73,74], and a shift from CD56bright (cytokine producing) NK to CD56dim (cytotoxic) NK cells[31,33,34]. Reports of regulatory T cell differences are mixed[28,31,75]. Although we did not observe systemic differences in our model, our findings corroborate dysregulated cytokine and immune cell recruitment with specificity to the maternal reproductive tissues and in response to an opportunistic bacterial pathogen. A limitation of our study is that we did not immune profile mock-infected control and GDM mice and it remains possible that there are baseline immune differences in GDM mice; however, cytokine levels were quite similar in mock-infected controls suggesting minimal altered reproductive immune function in the absence of infection. Another limitation is that immune responses were evaluated 72 h after GBS challenge, and thus more acute differences may have been missed. Characterizing other markers of aberrant inflammation such as immune cell sub-tissue localization and activation, and tissue damage, are key next steps.

Placentae from women with GDM have decreased regulatory T cells[76], and increased NK cells[33], neutrophil infiltration and activation[35], and macrophage activation[77]. Our data suggest that placental susceptibility to GBS is in part driven by impaired immune recruitment during infection. Other potential contributing factors include altered immune cell activation, compromised placental integrity, or aberrant iron storage, which have been reported in human GDM placentae[78–80]. We focused on two cell types based on comparisons between GDM and control hosts: neutrophils, highlighted by cytokine and host transcriptional differences, and natural killer cells, based on differing uterine proportions and placental counts in flow cytometry analyses.

With evidence of decreased uterine NK cells in GBS-challenged GDM mice, we hypothesized that NK depletion would worsen fetal infection. Surprisingly, NK depletion protected against GBS fetal invasion in GDM mice suggesting a deleterious role of NK cells during GBS infection in gestational diabetes. Although NK cells were systemically depleted, the observations that (1) infection was constrained to the reproductive tract in most mice (Fig. 1d), and (2) that the proportion of CD69+ NK cells was increased specifically in the uterus (Fig. 6e), together suggest observed effects were predominantly driven by uNKs, not peripheral NKs (pNKs). Murine and human uNKs are

proangiogenic and less cytotoxic compared to pNKs[81,82], constitutively express CD69 in mice[53], and display receptors for MHC I and MHC ligand recognition, enabling cooperation with antigen presenting cells[81,83]. Studies on uNKs in local bacterial infections in pregnancy are scarce;[83-85] one study on decidual NKs (dNKs) showed that human dNKs selectively transfer granulysin to kill intracellular *Listeria monocytogenes* in placental cells[86]. NK depletion in bacteremic non-pregnant mice improves survival against some bacterial species[87,88], worsens survival in some instances[89], and does not appear to impact GBS systemic infection[84,87]. GBS itself may manipulate NKs; GBS induces robust IFNγ production in peripheral NKs ex vivo[87] and engages Siglec-7, a CD33-related inhibitory receptor, to suppress NK inflammasome activation[90]. How NK anti-GBS activity is affected by the maternal-fetal tolerogenic environment of pregnancy, or diabetic pregnancy, are unknown. Here, we provide evidence that NKs display decreased activation in GDM hosts and that NK-induced inflammation may be detrimental to controlling GBS ascending fetoplacental infection. We speculate that NK depletion in GDM mice contributed to a reduced inflammatory state and improved control of GBS fetoplacental invasion, and that the beneficial impacts of reduced NK cells were muted in the less inflammatory controls.

Neutrophils and macrophages are recruited to gestational tissues and engage various antibacterial strategies to limit ascending GBS infection;[80,91,92] however, GBS may evade immune responses by dampening activation[93] or evasively hiding in macrophages[94]. Macrophage depletion decreases fetoplacental GBS burdens, but not maternal reproductive burdens, in a mouse model of ascending infection[94]. In this study, we are the first to report that, in addition to increasing vaginal GBS carriage, neutrophil depletion increases the frequency of GBS placental positivity without affecting fetal invasion, implying that other protective mechanisms constrain GBS in non-diabetic pregnancy. Neutrophil depletion had minimal impact on GBS pathogenesis in GDM mice, although it is possible that our study was underpowered to detect differences: we observed 40% incidence of preterm birth in neutrophil-depleted GDM dams and 0% in neutrophil-depleted controls but this difference was not significant. It is also possible that GDM mice are at the upper threshold or limit of GBS susceptibility, and thus it may not be biologically plausible to further aggravate GBS infection in our model.

In healthy human pregnancy, *Lactobacillus* spp. are enriched compared to non-pregnant or post-partum individuals and in general display more stability and lower alpha diversity[95-97]. Notably, the vaginal microbiome of pregnant mice is understudied despite the abundance of mouse models of obstetric disease. Three murine studies have characterized the maternal vaginal microbiota at discrete gestational timepoints and predominant vaginal taxa vary widely across studies[98-100]. This is the first longitudinal characterization of the vaginal microbiota in pregnant and diabetic pregnant mice, an important step in contextualizing the role of this bacterial community in reproductive health models. The dominant vaginal taxa in pregnant mice overlapped with non-pregnant mice from the same cohorts and from prior studies in non-pregnant C57BL/6J mice from Jackson labs[62,63,101] suggesting that pregnancy does not alter the overall community composition, although we did detect reduction in *S. succinus* mid-gestation. In pregnant controls, but not GDM mice, we observed increased alpha diversity as gestation progressed, which has been reported in some women with *L. iners* and *L. crispatus* dominant communities[96]. The vaginal microbiota in women with GDM is more diverse and enriched for dysbiotic genera including *Bacteroides, Klebsiella, Enterococcus*, and *Enterobacter*, and consequently alters neonatal microbial inheritance in oropharyngeal and intestinal tissues[37,38,60,61]. Similarly, we observed increased *Enterococcus* in vaginal tissue in GDM mice and increased co-ascension of *Enterococcus* with GBS into the uterine and/or fetoplacental tissues in a subset of mice. Whether *Enterococcus* is a bystander or contributing to pathogenesis of these tissues is currently unknown. We also observed that control

dams who experienced GBS invasion in offspring had reduced *Lactobacillus* during mid-gestation, while early gestation *Enterobacteriaceae* abundance increased in both GDM and controls that experienced perinatal invasive GBS. Together, these data demonstrate that this model recapitulates aspects of vaginal dysbiosis in gestational diabetic hosts and implicates specific taxa in influencing GBS pathogenesis in pregnancy. The functional roles of these taxa and whether differences in maternal microbiota composition instigate differences in neonatal microbial communities requires further investigation.

Our translational mouse model of GBS vertical transmission in gestational diabetic hosts recapitulates several clinical features and provides a foundation for further mechanistic and therapeutic exploration. Our findings highlight multifactorial drivers of GDM susceptibility to fetoplacental infection which include maternal immunity, pathogenic bacterial adaptations, and disruption of vaginal microbial changes in pregnancy. This preclinical model captures biological variability in GBS interactions with the gravid host that may aid in risk-stratification of women with GDM and provides a critical platform for testing alternative treatment options to improve perinatal outcomes in gestational diabetic pregnancy.

## Methods

### Ethics statement
All animal protocols and procedures were approved by the Baylor College of Medicine (BCM) Institutional Animal Care and Use Committee protocol AN-8233 and conducted under accepted veterinary standards and in compliance with all relevant ethical regulations. Biosafety level 2 work was performed under approval from the BCM Environmental Safety Committee.

### Bacterial strains, growth conditions and inoculation preparation
GBS strains A909 (ATCC BAA-1138) and CNCTC 10/84 (ATCC 49447), or isogenic A909 Δ*cylE*[51], or isogenic A909 Δ*yfhO*, were grown to stationary phase at 37 °C in Todd-Hewitt broth (Hardy Diagnostics) for at least 16 h. Cultures were diluted in fresh Todd-Hewitt broth and incubated at 37 °C until mid-logarithmic phase (defined as $OD_{600} = 0.4-0.6$). Bacterial cultures were centrifuged ($3220\,g$, 5 min), washed in sterile PBS, and resuspended at the desired concentration in sterile PBS.

### Construction of GBS *yfhO* mutant strain
To delete y*fhO* (SAK_RS10730, NCBI RefSeq accession NC_007432.1) in GBS A909, we used an allelic exchange approach as previously described[102]. Briefly, GBS A909 chromosomal DNA was used as a template for amplification of two 700 bp DNA fragments using two primers pairs: YfhO-BamH-F(5′-CGT CTG GAT CCC TGC ACT TAT TGG ACA AAA TG-3′)/Kan-YfhO-R1(5′-CAG TAT TTA AAG ATA CCG GTA TAC GAA GCT TAT AGT G-3′) and Kan-YhfO-R2(5′-TGA TGA AAG CCA TCG CGT ACT AAA ACA ATT CAT CCA G-3′)/YfhO-XhoI-R(5′-GTG CGC TCG AGC TAC ATA AAT CAT AGG AAT AGA GCC-3′). The designed primers contained 16−20 bp extensions complementary to the nonpolar kanamycin resistance cassette. The nonpolar kanamycin resistance cassette was PCR-amplified from pOSKAR (GenBank ID HM623914) using the primer pair YfhO-Kan-F1(5′-ATA AGC TTC GTA TAC CGG TAT CTT TAA ATA CTG TAG-3′)/YfhO-Kan-R2 which contained 16−20 bp extensions of complementary two DNA fragments of *yfhO*. The two fragments of *yfhO* and the fragment with the kanamycin resistance cassette were purified using the QIAquick PCR purification kit (Qiagen) and fused by Gibson Assembly (SGA-DNA) using primer pair YfhO-BamH-F/YfhO-XhoI-R. The assembled DNA fragment was digested with BamHI/XhoI and cloned into pHY304 digested with the respective enzymes. The plasmid was transformed into competent GBS A909 cells by electroporation and erythromycin resistant colonies were selected on THY agar plates at 30 °C. Integration was performed by growth of transformants at 37 °C with erythromycin selection. Excision

of the integrated plasmid was performed by serial passages in THY media at 30 °C and parallel screening for erythromycin-sensitive and kanamycin-resistant colonies. Double-crossover recombination was confirmed by PCR and Sanger sequencing using the following primer pair: YfhO-internal-F (5′-GGT TGG AAC AAA TAG TGT CC-3′) and YfhO-internal-R (5′-GAA TAA GCT GTT TGA ACC ATG-3′).

## Animals

Wild-type female and male C57BL/6J mice aged 6 weeks were purchased from Jackson Laboratories (strain code 000664). Male mice were used solely for mating and female mice were used for all in vivo experiments. Control and GDM groups were assigned randomly and mice were housed at 3 animals per cage. Mice ate and drank *ad libitum* and had a 12 h light cycle per day with housing conditions at maintained at 68–72 °F and 30–70% humidity. Timed mating was performed by introducing one male into a cage housing 3 females for a total of either two nights (pup outcomes experiments) or three nights (for *in utero* dissemination and immune profiling experiments). After each night, males were rotated between control and GDM cages to equalize exposure to productive male breeders. Soiled bedding from male cages was introduced into female cages three days before mating to synchronously induce estrus and promote successful mating. Embryonic day 0.5 (E0.5) of gestation was considered noon of the day after the first night of mating.

## In vivo model of GBS vaginal colonization in gestational diabetic mice

To establish gestational diabetes in mice, a diet-induced approach was used as previously described[40]. Briefly, mice are fed either a high-fat high-sucrose (HFHS) diet (D12451, Research Diets Inc.) or a control low-fat, no-sucrose diet (D12450K, Research Diets Inc.) one week before mating and maintained on the diet throughout pregnancy until the experimental endpoint. On days 14.5 and 15.5 of gestation, $1 \times 10^7$ colony forming units of GBS in a 10 µL PBS solution, was introduced into the vaginal tract with a gel loading pipette tip as described previously[103] to model mid-gestational vaginal colonization by GBS. Due to the multi-day mating schedule as described above, there is a range in day(s) of GBS challenge of E12.5-E15.5 for mice mated for 3 nights and E13.5-E15.5 for mice mated for two nights. Accordingly, the experimental endpoint of E17.5 has a +/-2 or +/-3 day window.

## Glucose tolerance test and tissue glucose measurement

For intraperitoneal glucose tolerance tests (IPGTT), mice were fasted for 4 h after which baseline fasting glucose levels were measured in venous tail sampling[40]. Subsequently, mice were given i.p. glucose (1 mg/g) and blood glucose levels were measured in duplicate at 30, 60, 90 and 120 min with two ReliOn Prime Blood Glucose Monitoring System meters (Walmart, Bentonville, AR). IPGTTs were performed one day before mating and on E13.5. Glucose levels in vaginal, uterine and placental tissues were measured in E17.5 tissue lysate samples with the Glucose Colorimetric kit (ThermoFisher, Cat. EIAGLUC) per manufacturer protocol.

## GBS in utero dissemination and pup outcomes analyses

To assess dissemination of GBS from the vaginal inoculation site upwards to the uterine-fetal space before birth, mice were inoculated with GBS as described above and then sacrificed on day E17.5, two days before expected delivery. Maternal (vagina, cervix, uterus), fetal (liver), and maternal-fetal (placenta) tissues were harvested. Placentae and livers were collected from each conceptus per dam, thereby permitting comprehensive tracking of GBS in utero dissemination and placental-fetal invasion. Dissected tissues were placed into 2 mL screwcap tubes containing 500 µL PBS and 1.0 mm zirconia/silica beads. The tubes were weighed and then kept on ice until homogenized in a Roche Magnalyser bead beater at 6000 rpm speed for

60 s. Ten-fold serial dilutions of tissue homogenates were plated on selective CHROMagar StrepB (DRG International, Inc.). Recovered GBS was identified as pink/mauve colonies and growth of blue colonies was considered endogenous *Enterococcus* spp. based on manufacturer protocols. Plates with bacterial lawns were recorded as having 2000 colonies, as this was deemed the upper limit of colony enumeration. Randomly selected blue colonies from vaginal, uterine and placental tissues were confirmed as *E. faecalis* by 16S rRNA gene sequencing. *In utero* litter features such as fetal intrauterine positions, fetal resorptions and total number of fetuses were also recorded. Dams with GBS detected in uterine and/or fetal samples were considered to have ascending infection. Offspring sex was determined by PCR with primers targeting *Rbm31x/y* F (5′-CAC CTT AAG AAC AAG CCA ATA CA-3′) and R (5′-GGC TTG TCC TGA AAA CAT TTG G-3′) as previously described[104]. Briefly, 10 µL of thawed pup intestinal or placental samples were diluted 1:100 in molecular grade $H_2O$, boiled at 95 °C for 15 min and then 2 µL of the resulting solution was used in a PCR reaction containing 1 µL forward primer, 1 µL reverse primer, 6 µL of $H_2O$, and 10 µL of 2X platinum Hotstart PCR mastermix. Samples were run in thermocycler with the following conditions: 94 °C for 2 min, followed by 30 cycles of 94 °C for 20 s, 50 °C for 20 s, 68 °C for 30 s, and then 68 °C for 5 min followed by a hold at 4 °C. The samples were then run on 1.3% agarose gel, containing SYBR Safe, at 120 V. Gels were visualized on a UV transilluminator.

In addition to characterizing the susceptibility of gestational diabetic mice and their fetuses to ascending GBS infection before birth, we also assessed pup outcomes during the first week of life in a separate mouse cohort. Mice were inoculated as described above, individually housed after E15.5 inoculation and then monitored twice daily for pre-term labor, pup birth and survival until postnatal day 7. On postnatal days 3 and 7, pups were weighed. Upon death or postnatal day 7 sacrifice, pup intestines and livers were harvested and processed as described above to quantify GBS burden. All tissue homogenates were stored at −20 °C until further use.

## Quantifying cytokines in vaginal, uterine and placental tissues

Vaginal, uterine, and placental tissue homogenates cytokines (IL-1α, IL-1β, IL-2, IL-3, IL-4, IL-5, IL-6, IL-9, IL-10, IL-12 (p40), IL-12 (p70), IL-13, IL-17A, Eotaxin, G-CSF, GM-CSF, IFN-γ, KC, MCP-1, MIP-1α, MIP-1β, RANTES, and TNF-α) were quantified via a 23-plex assay (cat. M60009RDP, Bio-Rad). Tissue samples were thawed on ice and then spun at 10,000 *g* for 10 min. Vaginal and uterine tissue supernatant were diluted 1:10 in Bio-Rad sample diluent, and placental tissues were diluted 1:2. Samples were then processed per the manufacturer's protocol. Data was acquired with Luminex xPONENT for Magpix, version 4.2 build 1324 on a Magpix instrument, and data was analyzed with Milliplex Analyst, (version 5.1.0.0).

## Immune cell profiling of vaginal, uterine and placental tissues

Vaginal and placental tissues were transected with one-third processed for burden quantification and the remaining two-thirds processed for flow cytometric quantification of immune cell populations. For uterine tissues, each horn was transected in half, and the halves from each horn were pooled and processed for either burden quantification as described above, or for immune cell quantification. Tissues harvested for flow were placed in 450 µL (placenta) or 900 µL (vagina, uterus) of RPMI, and mechanically disrupted with scissors until ~90% of sample was fine enough to pass through a p1000 tip. Collagenase (0.2 mg/mL) and DNase (50 µ/mL) were added to each vaginal and uterine sample, and Collagenase (0.2 mg/mL) and DNase (50 µ/mL) were added to placental samples. After vortexing, the samples were incubated at 37 °C and 250 rpm shaking for 30 min. 350 µL of supernatant was then filtered (40 µm) into an Eppendorf containing 800 µL RPMI + 10% FBS, and the resulting filtered cells were kept on ice. The remaining tissue fragments underwent a second digestion after supplementing with 350 µL total of

collagenase, DNase and RPMI at concentrations specified above per tissue type. After a second incubation at 37 °C and 250 rpm for 30 minutes, 350 µL of supernatant was filtered into the collection tubes containing single cells from previous filtration step. Samples were then spun at 500 × g for 10 min, resuspended in 500 µL of Red blood cell lysis buffer (Lucigen, SS000400-D2) and incubated at room temperature for 5 min. 700 µL of PBS was then added, samples were spun at 500 × g for 10 min, and then resuspended in 50 µL of PBS. Cells were then stained with 50 µL of Zombie Aqua that was reconstituted and diluted (1:1000 dilution) per manufacturer instructions. Cells were then incubated for 15 min at room temperature in the dark. Subsequently, cells were washed with 150 µL PBS, spun for 10 min and then resuspended in 50 µL of a 1:200 dilution of Fc block (CD16/CD32, clone 2.4G2, BD Biosciences, cat. 553141, 0.5 mg/mL) in FACS buffer (PBS, 1 mM EDTA, 1% FBS, 0.1% sodium azide), followed by a 15 min incubation at 4 °C in the dark. Cells were then incubated with an 18 antibody cocktail for 30 min, in the dark, at 4 °C. The antibody cocktail comprised: anti-CD3 (BV480, clone 145-2C11, cat. 746368, BD Biosciences), anti-CD45 (BV605, clone 30-F11, cat. 563053, BD Biosciences), anti-CD11b (APC-Cy7, clone M1/70, cat. 561039), anti-CD11c (BV785, clone N418, cat. 117335, BioLegend), anti-Ly6G (AF700, clone 1A8, cat. 561236, BD Biosciences), anti-NK1.1 (BUV737, clone PK136, cat. 741715, BD Biosciences), anti-CD19 (BB515, clone 1D3, cat. 564509, BD Biosciences), anti-CD8a (PE-Cy7, clone QA17A07, cat. 155018, BioLegend), anti-CD4 (BUV563, clone GK1.5, cat. 612923, BD Biosciences), anti-CD23 (BUV395, clone B3B4, cat. 740216, BD Biosciences), anti-CD44 (BUV805, clone IM7, cat. 741921, BD Biosciences), anti-CD64 (BV421, clone X54-5/7.1, cat. 139309, BioLegend), anti-MHCII (BV650, clone M5/114.15.2, cat. 563415, BD Biosciences), anti-CD24 (BV711, clone M1/69, cat. 563450, BD Biosciences), anti-CD62L (PE, clone M1/69, cat. 161204, BioLegend), anti-CD69 (PE-CF594, clone H1.2F3, cat. 562455, BD Biosciences), anti-Ly-6C (PerCP-Cy5.5, clone HK1.4, cat. 128012, BioLegend), anti-CD25 (APC, clone PC61, cat. 557192, BD Biosciences). Each antibody was added at 0.25 µL/sample for all antibodies except for anti-CD23, anti-CD24, and anti-Ly-6C for which 0.125 µL was added, and anti-CD3 for which 0.5 µL was added. Antibodies were combined with 10 µL brilliant stain buffer plus (cat. 566385, BD Biosciences) per sample and brought up to a total volume of 50 µL/sample with cell staining buffer (cat. 420201, BioLegend). After a 30 min incubation, samples were then washed with 100 µL of PBS, spun down (500 × g) for 10 min and then resuspended in about 300 µL FACS buffer. Data were acquired using a BD FACSymphony A5 and post-acquisition analyses were done using FlowJo software version 10.8. Gating strategy is shown in Fig. 4. Immune cell subsets were delineated from the CD45+ Zombie aqua⁻ population and defined based on the following staining profiles: basophils (CD19⁻, Ly6G⁻, MHCII⁻, CD11b⁺, CD62L⁻, Ly6c⁺/⁻ CD25⁻), dendritic cells (CD11b⁺, CD11C⁺/⁻, MHCII⁺, CD24⁺), eosinophils (CD19⁻, Ly6G⁻, MHCII⁻, CD11b⁻, CD62L⁻, Ly6c^lo CD25⁻), macrophages (CD11b⁺, CD11C⁺/⁻. MHCII⁺, CD24⁺/⁻ CD64⁺), mast cells (CD19⁻, Ly6G⁻, MHCII⁻, CD11b⁻, CD62L⁺/⁻, Cd11c^lo, SSCA^lo), monocytes (CD19⁻, Ly6G⁻, MHCII⁻, CD11b⁺, CD62L⁺), neutrophils (CD19⁻, Ly6G⁺), natural killer cells (CD19⁻, Ly6G⁻, NK1.1⁺), MHCII⁺ other cells (CD19⁻, Ly6G⁻, CD11b⁺, CD11c⁺, MHCII⁺, CD24L⁻ CD64L⁻), MHCII⁻ other cells (CD19⁻, Ly6G⁻, CD11b⁻, CD11c⁺, MHCII⁻, CD62L⁺/⁻), B cells (CD19⁺, Ly6G⁻), CD4⁺ T cells (CD19⁻, Ly6G⁻, CD11b⁻, CD11c⁻, CD4⁺), regulatory T cells (Tregs) CD4₊ T cells (CD19⁻, Ly6G⁻, CD11b⁻, CD11c⁻, CD4⁺, CD25⁺), active CD4⁺ T cells (Tregs) CD4₊ T cells (CD19⁻, Ly6G⁻, CD11b⁻, CD11c⁻, CD4⁺, CD69⁺), naïve CD4⁺ T cells (CD19⁻, Ly6G⁻, CD11b⁻, CD11c⁻, CD4⁺, CD62L⁺), memory CD4⁺ T cells (CD19⁻, Ly6G⁻, CD11b⁻, CD11c⁻, CD4⁺, CD44⁺), CD8⁺ T cells (CD19⁻, Ly6G⁻, CD11b⁻, CD11c⁻, CD8⁺), naïve CD8⁺ T cells (CD19⁻, Ly6G⁻, CD11b⁻, CD11c⁻, CD8⁺, CD62L⁺), memory CD4⁺ T cells (CD19⁻, Ly6G⁻, CD11b⁻, CD11c⁻, CD8⁺, CD44⁺).

## Immune cell depletion experiments

For NK cell depletion experiments, mice received intraperitoneal injections of 250 µg of anti-NK1.1 antibody (Bioxcell, clone PK136, cat. BE0036, Lot 828622A2), or 250 µg of mouse IgG2a isotype control (Bioxcell, clone C1.18.4, cat. BE0085, Lot 833922A2) on E13.5 and E15.5 in 100 µL. For neutrophil depletion experiments, mice received intraperitoneal injections of 200 µg of anti-Ly6G antibody (Bioxcell, clone 1A8, cat. BE0075-1, Lot 807722M1), or 200 µg of rat IgG2a isotype control (Bioxcell, clone 2A3, cat. BE0089, Lot 849322J2) on E13.5 and E15.5 in 100 µL. Mice were challenged with vaginal inoculations of GBS on E14.5 and E15.5 and sacrificed on E17.5 to assess in utero bacterial dissemination as described above. Uterine tissues were harvested and processed for flow cytometry as described above, with a few alterations in the work flow: single cells were fixed in 2% paraformaldehyde in RPMI for 10 min, washed with 500 µL RPMI, and then stained with live/dead, CD45, CD11b, NK1.1. or Ly6G antibodies overnight at 4 °C. Data were acquired and analyzed as described above to confirm targeted depletion.

## RNA-sequencing tissue processing and analysis

Vaginal and uterine tissues (E17.5) from GBS-challenged GDM and pregnant control mice were dissected and placed in tubes containing 500 µL RNA Protect and 1.0 mm zirconia/silica beads. Vaginal tissues were transected with half processed for RNA-sequencing and half processed for GBS quantification. Left uterine horns were processed for GBS quantification and right uterine horns were processed for RNA sequencing. Tissues were kept on ice until homogenized in a Roche Magnalyser bead beater at 6000 rpm speed for 60 s, spun at 13,000 rpm for 10 min. The resulting supernatant was discarded and pellets were resuspended in 100 µL TE and stored at −80 °C, until shipment to SeqCenter where RNA extraction, rRNA depletion and dual-RNA sequencing (50 million reads per sample) on an Illumina Stranded RNA-seq platform were performed. For bacterial sequences, quality control and adapter trimming was performed with bcl2fastq[105], read mapping was performed with HISAT2[106], and read quantification was performed using Subread's featureCounts[107], with alignment to the A909 reference genome (NCBI Nucleotide accession number NC_007432.1). For murine sequences, quality control and adapter trimming were performed with bcl-convert[108]. Read mapping was performed via STAR[109] to the mm10 version of the mouse genome and feature quantification was performed using RSEM[110]. Raw counts normalization and differential expression analyses were performed using R package DESeq2 (v 1.40.1)[111]. RStudio[112] (2022.12.0 + 353) was used to generate all heatmaps, PCA and volcano plots and pathways enrichment plots. The enhanced volcano[113] R package was used as well as ashr[114] LFC shrinkage for data visualization. The fGSEA R package (v1.20.0, RRID:SCR_020938) was used for gene set enrichment analyses with 10,000 permutations, a minimum gene set of 15, a maximum gene set of 500, and the Reactome gene set collections from the Molecular Signatures Database[115,116].

## YfhO structural predictions

AlphaFold2[45,46] was used to generate predictions from amino acid sequences for YfhO (Streptococcus agalactiae gene locus GL192_08955), transmembrane domains predicted by UniProt (A0A8I2JTU5)[117] and image generated using ChimeraX 1.6.1[118]. Homology detection and structure prediction were performed by the HHpred server[47].

## Longitudinal 16S rRNA sequencing of the maternal vaginal microbiota

To characterize the murine vaginal microbiota throughout healthy and gestational diabetic pregnancy, we serially sampled the vaginal lumen every three days beginning seven days before mating until day of GBS challenge (E14.5). Vaginal swabs were collected by gently rotating the swab 4 times clockwise and 4 times counter clockwise while applying slight pressure to the vaginal wall as described previously[103] for 3 out of 4 pup outcomes experiments and for one E17.5 in utero dissemination experiment. Non-pregnant cage mates were also swabbed every 3 days. For each swab collection, we included an environmental control

comprised of a swab exposed to the laminar hood air circulation for several seconds with subsequent submersion in sterile PBS. These PBS blanks were used as controls for environmental and sequencing contamination. Swab samples were then vortexed to dislodge biomaterial from the swab tips and the swabs were removed and discarded. The remaining PBS sample was then stored at −20 °C until further processing. Bacterial DNA was extracted from thawed PBS swab samples with the Quick-DNA Fungal/Bacterial Microprep kit following the manufacturer's protocol (Zymo Research) with final elution in 20 µL of water. The v4 region of the bacterial 16S rRNA gene was amplified by PCR using primer 515 F and 806 R and sequenced on an Illumina MiSeq v2 using the 2 × 250 bp paired-end protocol by the BCM Alkek Center for Metagenomic and Microbiome Research (CMMR). Raw data files were converted into FASTQs and demultiplexed using the Illumina 'bcl2fastq' software and single-index barcodes. Demultiplexed read pairs underwent initial quality filtering using bbduk.sh (BBMap version 38.82, 5) removing Illumina adapters, PhiX reads and reads with a Phred quality score below 15 and length below 100 bp after trimming. Quality controlled reads were merged using bbmerge.sh (BBMap version 38.82), and further filtered using vsearch[119] with a maximum expected error of 0.05, maximum length of 254 bp and minimum length of 252 bp. All the reads were then combined into a single fasta file by CMMR for further processing. We then joined and trimmed raw sequences to 150 bp, and denoised using Deblur through the DADA2 plugin on QIIME2 v2021.11 with taxonomic assignments determined using the naïve bayes sklearn classifier trained on the GreenGenes OTUs database (13_8, 99% sequence similarity). Dataset decontamination included filtering out OTUs that appeared in fewer than 5% of samples, removal of known DNA extraction contaminants[120,121], and processing samples through the Decontam R package[122]. After filtering, 40 total OTUs remained out of 3633 OTUs detected before filtering. Qiime2 was then used for diversity and compositional analyses of filtered, unrarefied, reads. Unless indicated by "other", all mapped OTUs are shown, and collapsed, at the genus level.

## Statistical analyses

Statistical analyses were performed using GraphPad Prism v9.5.1 as described in figure legends. Briefly, normality was assessed by the D'Agostino-Pearson normality test. Analysis of non-parametric data included two-tailed Mann-Whitney T tests for two groups, or Kruskal-Wallis test with correction for multiple comparisons for three or more groups. When appropriate, post-hoc Dunn's multiple comparisons test was utilized, or the two-stage linear step-up procedure of Benjamini, Krieger and Yekutieli to correct for multiple comparisons by controlling the false discovery rate (<0.05). Two-sided Fisher's exact tests were used for contingency analyses. The log-rank (Mantel–Cox) test was used to analyze survival curves, with Holm-Šídák correction for multiple comparisons when appropriate. For multiple paired samples (longitudinal Bray Curtis distance comparisons), analyses were conducted with Friedman's tests and Dunn's post hoc correction for multiple comparisons. DEGs were determined with a generalized linear model and genes with a $Log_2$ Fold Change >1 and FDR adjusted $p$ value < 0.05 were considered significantly different. For all experiments, a $p$ value of <0.05 was considered significant.

## Reporting summary

Further information on research design is available in the Nature Portfolio Reporting Summary linked to this article.

## Data availability

The 16 Sv4 rRNA sequencing data generated in this study have been deposited in NCBI Sequence Read Archive under BioProject accession number PRJNA988548. The A909 reference genome used in this study is publicly available on NCBI Nucleotide under accession number NC_007432.1 [https://www.ncbi.nlm.nih.gov/nuccore/NC_007432.1].

The RNA sequencing data generated in this study have been deposited in NCBI Gene Expression Omnibus under GEO accession number GSE236335 and BioProject accession number PRJNA990648. Other source data used to generate figures are provided with this paper as a Source Data file. Source data are provided with this paper.

## Code availability

The code is accessible at GitHub under project "Transcriptional-and-vaginal-microbial-analyses-in-a-mouse-model-of-gestational-diabetes-." [https://zenodo.org/records/10505424][123].

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

## Acknowledgements

This project was supported by the Cytometry and Cell Sorting Core at Baylor College of Medicine with funding from the CPRIT Core Facility Support Award (CPRIT-RP180672), the NIH (CA125123 and RR024574) and the assistance of Joel M. Sederstrom, Padmini Narayanan and Claude Chew. The Center for Metagenomics and Microbiology Research for 16 S sequencing (CMMR) and the Texas Medical Center Digestive Diseases Center (DDC), with funding from the NIH (DK056338), were also important resources for this study. This work was also supported in part by Cancer Prevention & Research Institute of Texas Proteomics & Metabolomics Core Facility Support Award (RP210227) and NCI Cancer Center Support Grant (P30CA125123) to the Antibody-based Proteomics Core/Shared Resource. We thank Susan F. Venable from the DDC and Shixia Huang, Zhongcheng Shi, Yuan Yao and Michael Nguyen from the Antibody-based Proteomics Core/Shared Resource for their excellent technical assistant in performing the Luminex experiments, data preliminary analyses and QC, and project consultation. We are also thankful to Misu A. Sanson-Iglesias and Luis A. Vega for helpful suggestions and resources for dual-RNA sequencing experiments and Anaid Reyes from the CMMR for assistance with 16S sequencing. Additionally, BioRender.com (2023) was used to generate experimental schematics. This work was supported by an NIH T32 award T32GM136554 to M.E.M. and J.J.Z. and NIH F31 award AI167547 to V.M.E., NIH F31 award AI167538 to M.E.M. and NIH F31 award HD111236 to SO. V.M.E. was also supported by a scholarship from Baylor Research Advocates for Student Scientists (BRASS) and a Grant for Emerging Researchers/Clinicians Mentorship Program from the Infectious Diseases Society of America (IDSA). Studies were supported by an R21 AI149366 to N.K., and a Burroughs Wellcome Fund Next Gen Pregnancy Initiative (NGP10103), NIH R01 (DK128053), U19 (AI157981), and R21 (AI173448) to K.A.P.

## Author contributions

KA Patras and VME conceived and designed experiments. VME, MEM, JJZ, SO, ZH, CS, CMR, MGM, MBB and SHR performed experiments. NK, KA Pennington, and ARF provided reagents. KA Patras and VME analyzed data and interpreted results. KA Patras, KA Pennington, and ARF secured funding. VME and KA Patras drafted the manuscript. All authors reviewed and edited the manuscript.

## Competing interests

The authors declare no competing interests.
