## [Peer Review File · Nature Communications]

Gestational diabetes augments group B *Streptococcus* infection by disrupting maternal immunity and the vaginal microbiotaREVIEWER COMMENTS

Reviewer #1 (Remarks to the Author):

This is the review for the manuscript Gestational diabetes augments group B Streptococcus perinatale infection through disruptions in maternal immunity and vaginal microbiota by Patras et al. Authors made an interesting study to understand host pathogen dynamics in a novel murine GDM model of GBS colonization and perinatal transmission using Dual RNA-Seq and 16s amplicon approaches. A thorough analysis was performed and well written methodologies, results and discussion. It is unclear if the authors used replicates for RNA-Seq approach and validation was done using RT-PCR approach. This would be more confident to the approach.

Reviewer #2 (Remarks to the Author):

This manuscript examines the relationship between gestational diabetes and adverse pregnancy outcomes in the setting of GBS colonization. This is common, complex clinical problem with serious implications for global maternal-child health. Using a newly developed murine model, the authors examine the impact gestational diabetes on GBS vaginal colonization, intrauterine infection and neonatal outcomes. Using dual-seq and flow cytometry/cytokine analysis they examine both pathogen and host responses in the diabetic milieu and used targeted GBS mutagenesis and immune depletion strategies to identify specific mechanisms underlying observed phenotypes. Finally, they performed longitudinal analysis of vaginal microbiome during pregnancy in both diabetics control mice and reveal specific microbial signatures associated with adverse pregnancy outcomes.

The major conclusion of these investigations, that gestational diabetes (in this specific animal model) is associated with greater GBS-related morbidities during pregnancy and early neonatal life- recapitulates what has been reported by some human studies and provides novel mechanistic pathways underlying these observations. Limitations of the model are acknowledged as is the complex interplay of diet, obesity, pregnancy and insulin sensitivity which complicate clinical/epidemiological studies.

This important work will be of great interest to the field and the readership of this journal. I have very few suggestions to enhance this strong manuscript.

Introduction:

Lines 63-64, 65-66: Recommend verbiage: "In the absence of intrapartum antibiotic prophylaxis administration, about half of infants born to GBS-positive women become colonized and a subset (~2%) proceed..." Clinical data support that GBS can invade the uterus prior to labor onset or still birth

Results:

Figure 1. Can the authors clarify what is meant by "placental containment" and how does it differ from the absence of GBS in fetal liver (ie differentiate between panels 1F and 1H)?

Why was Fisher's exact test chosen over Chi square test for data in Panels 1C,D,F and H (all cohorts have 5 or more)?

Figure 4. Can the authors speculate as to why the percentages of immune cells in the placenta relatively low versus that in the vaginal tissues? The placental compartment typically exhibits a robust inflammatory infiltrate in the setting of infection/inflammation.

Figure 7. Panels K-O: Please clarify the method used for Enterococcus detection (16s sequencing?). Did the authors examine GBS burden as well in these tissues using 16s and was it comparable to results obtained by using cultivation techniques.

Discussion: Line 498: Consider rewording statement that "NKs are detrimental to control GB S ascending fetoplacental infection" Perhaps NK induced inflammation is detrimental or some thing of the like.

Reviewer #3 (Remarks to the Author):

This article by Mercado-Evans et al. provides the most comprehensive evaluation to date of a mouse model of ascending GBS infection during pregnancy. In this paper, the authors adapted a diet induced approach to establish a gestational diabetes phenotype. Using a high-fat, high-sucrose diet 1 week prior to mating and during pregnancy, the authors then examined multiple facets of the host-pathogen interface looking for a mechanism to explain clinical data which suggests that mothers with GDM have increased vaginal colonization and poorer pregnancy outcomes. These investigations included looking within different reproductive tissue compartments (vagina, uterus, placenta) at bacterial burdens, microbial and host transcriptional profiles, cytokine profiles, immune cell populations (including NK and PMN depletion studies), neonatal outcomes, and microbiome evaluation. Probably not surprisingly, the authors found several differences between pregnant and non-pregnant mice (failed mating cohort) and control diet and HFHS diet (leading to GDM). Some of the most surprising results are the relatively minor changes in the groups despite the clear differences in outcomes. For instance, there were no differences in bacterial burden between the GDM and control groups in the uterus despite increased placental/fetal invasion, few differentially expressed genes in GBS were identified, modest changes in host tissue differentially expressed genes and cytokines existed, limited immune cell population changes (absolute or relative) were found, and microbiome changes were found but overall changes (alpha diversity) were not as different as one might have hypothesized. Overall, these numerous findings suggest a complicated web of multiple interacting factors that might contribute in specific settings (pregnant vs non-pregnant, GDM vs control) to lead to their outcome studies showing increased bacterial invasion and poorer neonatal health and survival. The authors took steps to further probe some of these possible mechanisms including creating an insertion-deletion mutant of one of the 22 differentially expressed microbial genes and PMN and NK cell depletion studies which add to the overall strength of the investigation.

These studies were meticulously conducted, and the controls appear appropriate for the experiments presented (including use of a different capsular type, hypervirulent GBS strain). This reviewer appreciates the clear labeling of data and the way the data are presented to maximize review of each data point in respect to the overall conclusion. The authors' conclusions are supported by the data presented and not overstated. Overall, I think this is an important paper to the field as animal models are the predominant method of study at present. While the investigation lacks a single, predominant mechanism to explain the model outcomes (in utero infection and neonatal outcomes), the totality of the descriptive work is still noteworthy and will be informative to the field.

This reviewer's primary critique is regarding the data pertaining to the model itself. The model's baseline characteristics which correspond to a 'GDM phenotype' (glucose intolerance, decreased serum insulin, insulin resistance, maternal dyslipidemia) are all cited from prior literature with no data presented verifying that the model is behaving as previously published during this study. To fully interpret the results of these several studies as within the context of "GDM", data is needed to ensure the HFHS diet does indeed lead to a consistent GDM phenotype. Aside from this major critique, the remainder of the critiques are relatively minor as the authors have done an outstanding job conducting and reporting a thorough investigation.

Major critique:

1. The authors refer to prior studies using this diet approach but provide no evidence of validation that the mice develop GDM characteristics in their hands. I would recommend a supplemental figure that shows at least changes in weight and glucose intolerance (fasting or glucose tolerance test) prior to and during pregnancy to demonstrate that the mice on the different diets are behaving like the cited studies.

Minor Critiques

2. If there was a range of glucose values within the GDM group, do the mice with the higher serum glucose levels have higher vaginal/uterine burdens? Is there any correlation between the extent of glucose levels and infection outcomes?
3. Were there any absolute changes in the vaginal microbiota (Figure 7/S5)? Only relative abundance data is shown.

Response to Reviewers

Reviewer #1 (Remarks to the Author):

This is the review for the manuscript Gestational diabetes augments group B Streptococcus perinatal infection through disruptions in maternal immunity and vaginal microbiota by Patras et al. Authors made an interesting study to understand host pathogen dynamics in a novel murine GDM model of GBS colonization and perinatal transmission using Dual RNA-Seq and 16s amplicon approaches. A thorough analysis was performed and well written methodologies, results and discussion. It is unclear if the authors used replicates for RNA-Seq approach and validation was done using RT-PCR approach. This would be more confident to the approach.

We thank the reviewer for their feedback. In our initial submission, we included 4 biological replicates per group for RNA-Seq; vaginal and uterine tissues were sequenced from 4 control and 4 GDM mice. While we did not validate using RT-PCR, we have since RNA-sequenced reproductive tissues from 8 additional mice (4/group) to validate our findings and ensure technical replicates in our study. All host and GBS RNA-sequencing data presented in the manuscript (**Fig. 2, Fig. 3, Supp. Table 1 and Supp. Table 2**) are updated to include the technical replicates. We also incorporated placental tissues findings to begin to clarify GDM-associated perturbations. For GBS uterine vs. vaginal comparisons, we opted to include a list of DEGs identified from the first technical replicate as this was how we first identified *yfhO*. We believe that this second round of RNA-sequencing, paired with the *yfhO in vivo* findings provides validation of the genes identified in our RNA-seq.

Reviewer #2 (Remarks to the Author):

This manuscript examines the relationship between gestational diabetes and adverse pregnancy outcomes in the setting of GBS colonization. This is common, complex clinical problem with serious implications for global maternal-child health. Using a newly developed murine model, the authors examine the impact gestational diabetes on GBS vaginal colonization, intrauterine infection and neonatal outcomes. Using dual-seq and flow cytometry/cytokine analysis they examine both pathogen and host responses in the diabetic milieu and used targeted GBS mutagenesis and immune depletion strategies to identify specific mechanisms underlying observed phenotypes. Finally, they performed longitudinal analysis of vaginal microbiome during pregnancy in both diabetics control mice and reveal specific microbial signatures associated with adverse pregnancy outcomes.

The major conclusion of these investigations, that gestational diabetes (in this specific animal model) is associated with greater GBS-related morbidities during pregnancy and early neonatal life- recapitulates what is has been reported by some human studies and provides novel mechanistic pathways underlying these observations. Limitations of the model are acknowledged as is the complex interplay of diet, obesity, pregnancy and insulin sensitivity which complicate clinical/epidemiological studies.

This important work will be of great interest to the field and the readership of this journal.

I have very few suggestions to enhance this strong manuscript.

Introduction:

Lines 63-64, 65-66: Recommend verbiage: “In the absence of intrapartum antibiotic prophylaxis administration, about half of infants born to GBS-positive women become colonized and a subset (~2%) proceed...” Clinical data support that GBS can invade the uterus prior to labor onset or still birth

We thank the reviewer for the overall positive and critical feedback that will improve our work. We have re-worded this section to clarify the prevalence of invasive disease.

Results:

Figure 1. Can the authors clarify what is meant by “placental containment” and how does it differ from the absence of GBS in fetal liver (ie differentiate between panels 1F and 1H)?

Absence of GBS in fetal liver (previously Fig 1F, now Fig 1G) included 2 scenarios: 1) GBS did not make it to fetal liver in some mice because it never successfully made it the placenta and instead was limited to the uterus or lower reproductive tract, or 2) GBS did not successfully make it to the liver because it became restricted to placental tissue.

To indirectly probe GDM-mediated effects on placental integrity we did a sub-analysis where we compared GBS progression to fetal livers once it has already reached placental tissues. To do this, we selected only GBS+ placentae and then assessed what fraction of these had GBS detected in their respective fetal liver unit. We hypothesized that GDM disrupts placental integrity and thus would be more permissive of GBS dissemination to fetal tissues. This is now clarified in the text (Pages 7-8, Lines 160-164).

Why was Fisher’s exact test chosen over Chi square test for data in Panels 1C,D,F and H (all cohorts have 5 or more)?

We made this decision based on our relatively small sample size. While the Chi square test relies on an approximation, Fisher’s exact test is one of exact tests – thus tends to be more accurate than Chi square when the sample size or number of expected events is small (PMID: 28503482). Though, in response to this point, we performed a Chi square and the findings are consistent.

Figure 4. Can the authors speculate as to why the percentages of immune cells in the placenta relatively low versus that in the vaginal tissues? The placental compartment typically exhibits a robust inflammatory infiltrate in the setting of infection/inflammation.

Revised Fig. 6B shows that a lower percentage of placental cells were CD45+ compared to vaginal cells; however, Supp. Fig. 6B demonstrates more total cells were collected from placental and uterine compared to vaginal tissues. This in part is due to larger mass of the tissues, but also more efficient tissue digestion. Further breakdown of CD45+ counts in Supp. Fig. 6C show that comparable levels of immune cells were counted between placental and vaginal samples (~ 1×10^4 cells). We performed an additional RNA sequencing experiment and included placentae which did indeed confirm upregulation of several inflammatory pathways (see revised Fig. 4D,G). We also expanded our cytokine analysis to include more placentae and this confirmed robust cytokine responses to GBS (see revised Fig. 5 and revised Supp. Fig. 3 and 4).

Figure 7. Panels K-O: Please clarify the method used for *Enterococcus* detection (16s sequencing?). Did the authors examine GBS burden as well in these tissues using 16s and was it comparable to results obtained by using cultivation techniques.

Enterococcus was detected by 16S for the E2 timepoints (revised Fig. 9O and Supp. Fig. 6n). For *Enterococcus* ascension (revised Fig.9P and Supp. Fig. 2C), *Enterococcus* was detected as differential (blue) colonies grown on the same medium used to quantify GBS (pink) colonies. This has been clarified in the text (Page 19 Lines 399-403, and Page 34, Lines 708-710) and figure legends. We did not examine GBS burden after inoculation in these tissues using 16S for a couple of reasons; 1) We did not want to influence pregnancy outcomes (preterm birth) via vaginal swabs/handling after bacterial challenge, and 2) we have previously observed that GBS often overtakes the vaginal microbiota such that community state types are shifted (PMIDs: 34986315, 30477439). During revision experiments (see below), we swabbed a few mice to confirm *Enterococcus* observations via cultivation and see a trend towards increased early *Enterococcus* in GDM but not controls, consistent with 16S findings.

Discussion: Line 498: Consider rewording statement that “NKs are detrimental to control GB S ascending fetoplacental infection” Perhaps NK induced inflammation is detrimental or some thing of the like.

We thank the reviewer for this excellent point. The text now reads: “...we provide evidence that NKs display decreased activation in GDM hosts and that NK-induced inflammation may be detrimental to controlling GBS ascending fetoplacental infection.” (Page 26, Lines 547-549).

Reviewer #3 (Remarks to the Author):

This article by Mercado-Evans et al. provides the most comprehensive evaluation to date of a mouse model of ascending GBS infection during pregnancy. In this paper, the authors adapted a diet induced approach to establish a gestational diabetes phenotype. Using a high-fat, high-sucrose diet 1 week prior to mating and during pregnancy, the authors then examined multiple facets of the host-pathogen interface looking for a mechanism to explain clinical data which suggests that mothers with GDM have increased vaginal colonization and poorer pregnancy outcomes. These investigations included looking within different reproductive tissue compartments (vagina, uterus, placenta) at bacterial burdens, microbial and host transcriptional profiles, cytokine profiles, immune cell populations (including NK and PMN depletion studies),

neonatal outcomes, and microbiome evaluation. Probably not surprisingly, the authors found several differences between pregnant and non-pregnant mice (failed mating cohort) and control diet and HFHS diet (leading to GDM). Some of the most surprising results are the relatively minor changes in the groups despite the clear differences in outcomes. For instance, there were no differences in bacterial burden between the GDM and control groups in the uterus despite increased placental/fetal invasion, few differentially expressed genes in GBS were identified, modest changes in host tissue differentially expressed genes and cytokines existed, limited immune cell population changes (absolute or relative) were found, and microbiome changes were found but overall changes (alpha diversity) were not as different as one might have hypothesized. Overall, these numerous findings suggest a complicated web of multiple interacting factors that might contribute in specific settings (pregnant vs non-pregnant, GDM vs control) to lead to their outcome studies showing increased bacterial invasion and poorer neonatal health and survival. The authors took steps to further probe some of these possible mechanisms including creating an insertion-deletion mutant of one of the 22 differentially expressed microbial genes and PMN and NK cell depletion studies which add to the overall strength of the investigation.

These studies were meticulously conducted, and the controls appear appropriate for the experiments presented (including use of a different capsular type, hypervirulent GBS strain). This reviewer appreciates the clear labeling of data and the way the data are presented to maximize review of each data point in respect to the overall conclusion. The authors' conclusions are supported by the data presented and not overstated. Overall, I think this is an important paper to the field as animal models are the predominant method of study at present. While the investigation lacks a single, predominant mechanism to explain the model outcomes (in utero infection and neonatal outcomes), the totality of the descriptive work is still noteworthy and will be informative to the field.

This reviewer's primary critique is regarding the data pertaining to the model itself. The model's baseline characteristics which correspond to a 'GDM phenotype' (glucose intolerance, decreased serum insulin, insulin resistance, maternal dyslipidemia) are all cited from prior literature with no data presented verifying that the model is behaving as previously published during this study. To fully interpret the results of these several studies as within the context of "GDM", data is needed to ensure the HFHS diet does indeed lead to a consistent GDM phenotype. Aside from this major critique, the remainder of the critiques are relatively minor as the authors have done an outstanding job conducting and reporting a thorough investigation.

Major critique:

1. The authors refer to prior studies using this diet approach but provide no evidence of validation that the mice develop GDM characteristics in their hands. I would recommend a supplemental figure that shows at least changes in weight and glucose intolerance (fasting or glucose tolerance test) prior to and during pregnancy to demonstrate that the mice on the different diets are behaving like the cited studies.

We are grateful for this reviewer's critical assessment of our study. We performed GTT experiments and observed glucose intolerance solely in the pregnant mice on the high fat high sucrose diet on E13.5 (revised Fig. 1B) and not in non-pregnant mice on the HSHF diet (revised Supp. Fig. 1A). Maternal weight was similar between groups as reported previously (Supp. Fig.

1B, PMID: 28203773). We also tested whether reproductive tissue glucose levels were different between GDM and control mice, but did not detect differences at the time of tissue collection (E17.5, Supp. Fig. 1C). These have been added to the text (Pages 6-7, Lines 133-144).

Minor Critiques

2. If there was a range of glucose values within the GDM group, do the mice with the higher serum glucose levels have higher vaginal/uterine burdens? Is there any correlation between the extent of glucose levels and infection outcomes?

This was a great suggestion. GTT area under the curve on E13.5 modestly trends with uterine burdens on E17.5 (lower right panel), but we did not see any significant correlations between tissue burdens and blood glucose on E13.5, nor tissue glucose concentrations on E17.5.

3. Were there any absolute changes in the vaginal microbiota (Figure 7/S5)? Only relative abundance data is shown.

This was another helpful suggestion. Now presented in Supp. Fig. 6E-F, we show that for both GDM and controls, there was a decrease in biomass reflected by fewer total mapped reads in pregnant mice (both control and GDM) but this was absent in non-pregnant mice. This has also been added to the text (Pages 17-18, Lines 372-374).

REVIEWERS' COMMENTS

Reviewer #1 (Remarks to the Author):

The authors clarified the comments in for the review regarding replicates and validation of results were addressed. My decision is to accept the article.

Reviewer #2 (Remarks to the Author):

The authors addressed all comments and suggestions thoroughly and thoughtfully. I have no further recommendations for edits.

Reviewer #3 (Remarks to the Author):

My prior concerns have all been adequately addressed. This was already a thorough and strong manuscript and has been strengthened by the additional revisions.